



# Using Deep Learning and Multi-source Remote Sensing Images to Map Landlocked Lakes in Antarctica

Anyao Jiang[1#], Xin Meng[1#], Yan Huang[1], Guitao Shi[1,2*]

[1] Key Laboratory of Geographic Information Science (Ministry of Education), School of Geographic Sciences and State Key
5  Laboratory of Estuarine and Coastal Research, East China Normal University, Shanghai, 200241, China
[2] Key Laboratory of Spatial-temporal Big Data Analysis and Application of Natural Resources in Megacities, Ministry of
Natural Resources, Shanghai, 200241, China
[#] These authors contributed equally to this work.

*Correspondence to*: Guitao Shi (gtshi@geo.ecnu.edu.cn)





**Abstract.** Antarctic landlocked lakes' open water (LLOW) plays an important role in the Antarctic ecosystem and serves as a reliable climate indicator. However, since field surveys are currently the main method to study Antarctic landlocked lakes, the spatial and temporal distribution of landlocked lakes across Antarctica remains understudied. We first developed an automated detection workflow for Antarctic LLOW using deep learning and multi-source satellite images. The U-Net model and LLOW identification model achieved average Kappa values of 0.85 and 0.62 on testing datasets respectively, demonstrating strong spatio-temporal robustness across various study areas. We chose four typical ice-free areas located along the coastal Antarctica as our study areas. After applying our LLOW identification model to a total of 79 Landsat 8-9 images and 390 Sentinel-1 images in these four regions, we generated high spatiotemporal resolution LLOW time series from January to April between 2017 and 2021. We analyzed the fluctuation of LLOW areas in the four study areas, and found that during expansion of LLOW, over 90 % of the changes were explained by positive degree days; while during contraction, air temperature changes accounted for more than 50 % of the LLOW area fluctuations. It is shown that our model can provide long-term LLOW series products that help us better understand how lakes change under a changing climate in the future.





## 1. Introduction

Antarctic lakes play a crucial role in the ecosystem of Antarctica and are reliable indicators of climate change (Lyons et al., 2006). These lakes can be divided into three main types: landlocked lakes, epiglacial lakes, and supraglacial lakes. Landlocked lakes, located in local depressions and usually free of ice during austral summer, primarily receive water inflow from the melting of seasonal snow cover (Shevnina et al., 2021). Epiglacial lakes are situated at the boundary between areas of rock and ice, and melting of the glacier ice is the main source of water inflow into them. Supraglacial lakes are found on the surface of ice sheets, glaciers, and ice shelves, forming during the summer melt (Hodgson, 2012).

Extensive research has confirmed the presence of microorganisms in Antarctic lakes (Parnikoza and Kozeretska, 2019). Simple and short food webs are the major characteristics of Antarctic lakes, mainly include prokaryotes (e.g., bacteria) and eukaryotes (e.g., phytoplankton) (Izaguirre et al., 2021). Taton et al. (2006) investigated cyanobacterial diversity and distribution in microbial mats of East Antarctic lakes, revealing their crucial role in the ecosystem. This finding aligns with the study by Komárek et al. (2012), who identified cyanobacteria as dominant contributors to primary production and nutrient cycling on James Ross Island. Additionally, Huang et al. (2014) examined bacterial communities, while Koo et al. (2014) isolated Hymenobacter sp. Strain IS2118 in landlocked lakes of Schirmacher Oasis. These investigations revealed the presence of unique microbial assemblages, highlighting their ecological importance in the lake environments. Moreover, Carvalho et al. (2008) investigated genetic diversity in Legionellaceae bacteria in landlocked lakes of Keller Peninsula, while Papale et al. (2017) explored prokaryotic communities in Lake Limnopolar, Byers Peninsula, providing insights into bacterial adaptation and evolution in isolated Antarctic lake ecosystems. Eukaryotes are detected in Antarctic lakes as well, such as phytoplankton and plankton (Keskitalo et al., 2013; Rochera and Camacho, 2019). These previous studies suggest that Antarctic lakes feature a very diverse set of ecosystems.

Antarctic lakes are rather sensitive to environmental shifts, especially under a warming climate (Quayle Wendy et al., 2002). Seasonally ice-covered lakes magnify the warming trends observed in air temperature (Convey and Peck, 2019). Recent studies have highlighted the impact of increased temperature and melting of snowfields and glaciers on Antarctic lakes (Izaguirre et al., 2021; Stokes et al., 2019). In particular, the changes in the lake-ice and open water area can have significant implications for the lake environment, affecting both physical and biological aspects. Physically, alterations in lake-ice and open water area influence thermal stratification, leading to variations in heat distribution and vertical mixing within the water column (Preston et al., 2016; Lazhu et al., 2021). This, in turn, has implications for the biological effects observed. The occurrence peak of primary consumers (Hébert et al., 2021; Izaguirre et al., 2021), nutrient regime (Prater et al., 2022; Yang et al., 2021), the development of planktonic and benthic microbial population (Camacho, 2006), and the availability of suitable oxythermal habitat for cold-water organisms (Pöysä, 2022) all can be influenced by the changes in lake-ice and open water area. Rising temperatures and stratification, coupled with reduced ice cover and increased nutrient inputs, may promote the growth of specific phytoplankton (Prowse et al., 2011). Landlocked lakes situated in coastal Antarctica typically undergo rapid species replacements during the active phytoplankton growth season, resulting in changes in plankton



abundance (Izaguirre et al., 2021). For example, observations in Lake Limnopolar, Byers Peninsula, have demonstrated that temperature-induced warming significantly alters carbon flow, thereby impacting the abundance of plankton in the lake

ecosystem (Villaescusa et al., 2016).

Over the past decade, thanks to the development of satellite remote sensing, there has been an increasing interest in the detection of Antarctic lakes. Compared to manual digitizing, automated lake detection method is more suitable for larger-scale assessments because it can be rapidly applied to hundreds of satellite scenes and can avoid user bias (Arthur et al., 2020). A number of methods have been developed to map Antarctic supraglacial lakes including threshold-based lake

classification methods (Fitzpatrick et al., 2014; Moussavi et al., 2020), adaptive classification methods (Johansson and Brown, 2013), and machine learning algorithms (Dirscherl et al., 2020, 2021a). Most of previous works mainly focus on the detection of supraglacial lakes (Dirscherl et al., 2021a; Dirscherl et al., 2021b; Leeson et al., 2015; Li Qing, 2021; Moussavi et al., 2020). Currently, the semi-automated algorithm has been developed for the detection of water bodies in Greenland (Miles et al., 2017). This method utilized Sentinel-1 Synthetic Aperture Radar (SAR) and Landsat 8-9 Operational Land

Imager (OLI) imagery to monitor surface and subsurface lakes on the Greenland Ice Sheet. As for Antarctic landlocked lakes, field surveys served as the primary method (Shevnina et al., 2021; Lecomte et al., 2016; Shevnina and Kourzeneva, 2017; Harris and Burton, 2010). Due to the limited study area scope and non-uniform of field surveys, the spatio-temporal distribution of landlocked lakes across Antarctica remains understudied. Thus, to better understand the dynamics of landlocked lakes in Antarctica, more efficient and accurate methods are needed.

This study aims to test the application of a deep learning approach to detect the landlocked lakes' open water (LLOW) area in Antarctica in combining the Landsat 8-9 OLI and SAR imagery. Then, we aim to investigate the variations in LLOW and their relationship with environmental factors, such as temperature. To the best of our knowledge, this study represents the first attempt to map the open water area of landlocked lakes in Antarctica using remote sensing data.

**2. Research Data**

**2.1 Study Area**

Four typical ice-free areas distributed on coastal Antarctica were selected as study areas (Fig. 1). Antarctic Peninsula have experienced the largest increases in near-surface air temperature in the Southern Hemisphere during the past decades (Turner et al., 2016). As a representative site of the Antarctic Peninsula, Clearwater Mesa (CWM; 57.71° W, 64.03° S) on James

Ross Island was chosen due to its high density of lakes, unique geomorphological setting, remote elevated position, and lack of previous human presence (Roman et al., 2019). In East Antarctica, we selected two large ice-free oases, Larsemann Hills (LHs; 76.23° E, 69.41° S) and Vestfold Hills (VHs; 78.18° E, 68.58° S). The VHs is a 400 km$^2$ area of ice-free rock (Seppelt and Broady, 1988), while LHs is the second largest ice-free oasis along the East Antarctica with an area of about 50 km$^2$ (Shi et al., 2018). The Schirmacher Oasis (SO; 11.65° E, 70.76° S), which is an east-west trending narrow strip, with an ice-free





area of about 35 km$^2$ (Srivastava et al., 2013), was chosen to represent the higher latitude regions of Antarctica. Since SO is located about 100 km from the coast, it can also represent the inland region of Antarctica. In these areas, the water of landlocked lakes is mainly from the melting of seasonal snow cover.

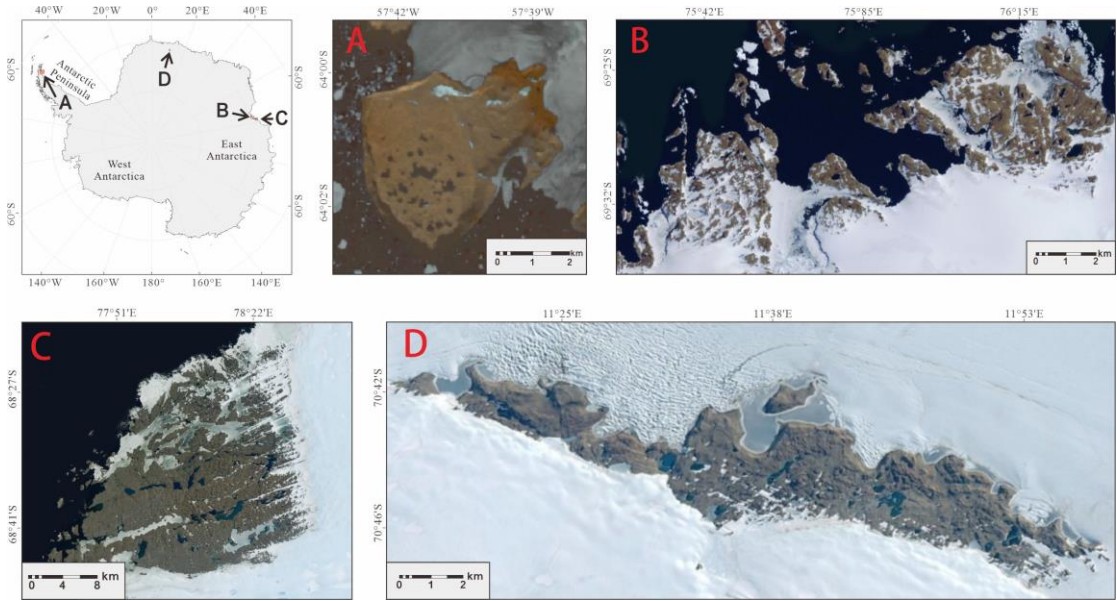

**Figure 1. Map of study areas. Satellite images based on Landsat 8 and Esri World Imagery scenes show examples of landlocked lake occurrence. Scenes used for this figure include: (A) Clearwater Mesa (CWM; Landsat 8; 2 February 2016), (B) Larsemann Hills (LHs; Esri World Imagery; 7 April 2022), (C) Vestfold Hills (VHs; Esri World Imagery; 7 April 2022), and (D) Schirmacher Oasis (SO; Esri World Imagery; 7 April 2022).**

**2.2 Dataset**

The OLI onboard the Landsat 8-9 satellite captures optical information in the visible, near infrared, and shortwave infrared portions (VNIR, NIR and SWIR), enabling the comprehensive assessment of diverse surface features. The Landsat 8-9 OLI data are superior in many aspects, such as the enhanced radiometric capabilities and the expanded range of spectral bands (Gorji et al., 2020). Leveraging the capabilities of Landsat 8-9 OLI facilitates the better monitoring of the LLOW. Thus, a

total of 79 optical images of Landsat 8-9 Collection 1 with 30 m resolution between January and April from 2014 to 2022 were obtained from the United States Geological Survey (USGS) Global Visualization Viewer (GLOVIS) portal (http://earthexplorer.usgs.gov/). Landsat 8-9 satellite has a 16-day repeat cycle. However, cloud cover frequently hampers the detection through visible bands within the study areas. Whenever thick layers of clouds are present above our study areas in the Landsat images, those images are excluded from our study. As a result, the time interval between usable Landsat

images can vary. In the Landsat OLI products, the optical bands 1-7 were utilized to identify the land cover in the study areas.



The Sentinel-1 mission is dedicated to SAR imaging and provides the all-weather, day-and-night imagery at C-band. The SAR-based landscape detection offers a distinct advantage over optical approaches by mitigating the challenges posed by cloud interference (Zhang et al., 2020). Consequently, it can offer important datasets for acquiring the long time-series monitoring of the LLOW. Because of the advantages of SAR images, Sentinel-1 datasets had been widely used for Antarctic

open water and snowmelt detection studies (Bowden et al., 2006; Liang et al., 2021; Dirscherl et al., 2021b). European Space Agency (ESA) facilitates access to various Sentinel-1 products, including raw level-0 data, processed level-1 Single Look Complex (SLC) data and level-1 Ground Range Detected (GRD) data. Considering the high temporal resolution requirement of LLOW detection tasks, we used a total of 390 high-resolution Sentinel-1 SAR images from the Interferometric Wide (IW) Swath GRD products with about 10-m pixel space, which were acquired from Alaska Satellite Facility (ASF)

(https://search.asf.alaska.edu/). These selected Sentinel-1 images for CWM, LHs and VHs span from 2017 to 2021. However, for the SO region, where Sentinel-1 images are unavailable prior to 2019, only the images during 2020 and 2021 were obtained. The revisit period of Sentinel-1 satellites is 12 days. By utilizing both Sentinel-1A and Sentinel-1B images, we obtained a shorter time interval of 6 days between consecutive Sentinel-1 images. These GRD products play a critical role in distinguishing the LLOW in the study areas.

Our dataset of daily mean near-surface temperatures for CWM and SO came from ERA5-land dataset obtained from Google Earth Engine (Muñoz Sabater, 2019). The daily mean air temperatures for LHs and VHs were derived from the weather stations at Zhongshan Station (Ding et al., 2022) and Davis Station. After, we used "temperature" to express "daily mean air temperature" and "daily mean near-surface temperature" for short. To facilitate the terrain correction, we employed the Copernicus 90 m Global DEM data.

During the Antarctic summer, the snow cover on the lake surface undergoes melting, and consequently the LLOW will be present, which can be easily observed through remote sensing techniques. The melting and freezing processes typically occur between September and April. However, the identification of LLOW is challenged by rising temperature events during September and December. These occasional temperature increases trigger the relatively high temperatures, and increased snow wetness. This wetness increase can diminish the backscatter of the snow surface (Shokr and Dabboor, 2020). The

events lead to the underestimation of both snow and snow-covered ice backscatter, resulting in similar backscatter characteristics among ice, snow, and LLOW. During the melting period of landlocked lakes, which spans from September to December, frozen landlocked lakes may be covered by wet snow due to rising temperature events, resulting in low backscatter. Consequently, these frozen lakes are not LLOW but incorrectly identified as LLOW. During January and April, the melt landlocked lakes have less snow cover and are less affected by the rising temperature events. Thus, the

identification of LLOW from January to April is much more accurate compared to September to December. To evaluate the influence of rising temperature events from September to December, we sampled pixels of open water, land ice layers and sea ice layers from several SAR images during this period. We also sampled LLOW pixels in January as the reference for backscattering analysis. We found that the backscatter of sampled land ice layers was as low as that of sampled LLOW in January in our study regions (Supplementary Table 1). Consequently, our model cannot effectively distinguish between



LLOW and ice layer in these images from September to December. In addition, compared to single-polarization SAR images, the utilization of multi-polarization SAR images can improve the capability to distinguish LLOW from other ground objects (Zakhvatkina et al., 2019), which mitigates the overestimation of LLOW caused by rising temperature events. However, the high-resolution GRD products only provide single polarization over the Antarctic continent. The high-resolution multi-polarization SAR images are not available in Antarctica; therefore, we cannot mitigate the LLOW identification errors from

September to December. Considering these factors, our analysis focuses on the changes in LLOW from January to April, when the identification accuracy is comparatively high.

## 3. Lake open water identification

The automated detection workflow for LLOW can be divided into three distinct steps (Fig. 2): (1) pre-processing of Landsat

and Sentinel input images, (2) open water identification, and (3) post-processing of extracted open water to generate the LLOW time series. To assess the accuracy of our LLOW detection workflow, we conducted a comparison between the identified LLOW and the labeled ground truth.





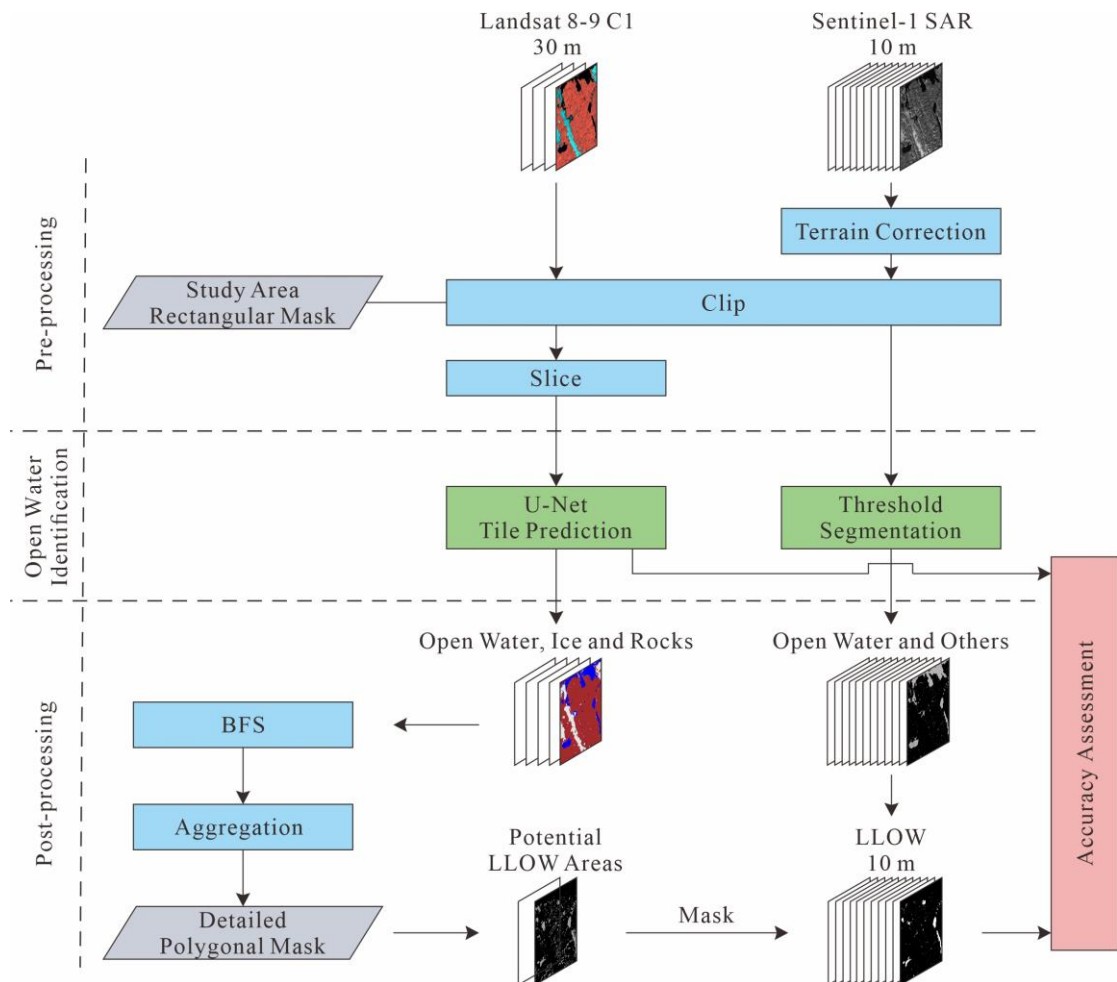

**Figure 2. Workflow for detecting the landlocked lakes' open water (LLOW) in Antarctica.**

## 3.1 Pre-Processing

The relative location of the study area in each image may vary across different satellite images, due to variations in satellite orbit tracks and subtle drifts within the same orbit track. To maintain consistency in the relative location of LLOW in each 165 image, we performed the terrain correction process and subsequently cropped images according to the specific study area boundaries. This approach ensures the uniformity and accuracy of the relative location of the study area. To optimize computational efficiency, the cropped Landsat images were further divided into multiple tiles using an overlap-tile strategy before input into the U-Net model. In addition, training and validation datasets were generated and augmented from the pre-processed Landsat images to train the U-Net model.





Ensuring consistent relative location of the study area in each image enhances the comparability of the detected LLOW within the study area across different images. To achieve this, the predefined rectangular boundaries were established based on projected coordinates. We cropped images to fit within these specific boundaries, thereby unifying the relative location of the study area in the predefined boundaries. For Landsat images, we utilized the specified coordinates to apply the resampling technique with Nearest Neighbor (NN) algorithm and perform image cropping. For Sentinel-1 images, we

merged the images and performed terrain correction on the Sentinel-1 Level-1 GRD products using ESA's Sentinel Applications Platform (SNAP) software. Then the corrected Sentinel-1 images were then reprojected and cropped to align with the spatial extent of the cropped Landsat images.

The use of an overlap-tile strategy for splitting large images into smaller patches has proven effective in overcoming GPU limitations (Ronneberger et al., 2015). Thus, this strategy was employed before using Landsat images in the U-Net. We

found that the trained U-Net is not ideal for recognizing small-scale open water, so we sliced the input images to the patches of 300*300 pixels, the common patch size in existing U-Net models. Then we performed resampling with NN to resize these patches of 300*300 pixels to 1024*1024 pixels, in order to magnify the small open water area. During tiling, the void input data on the edge is filled using a mirroring strategy. After land-cover classification using U-Net, we again resampled these classified results of 1024*1024 pixels to 300*300 pixels with NN. To avoid the deterioration of land-cover classification

caused by the edge of the slice, we only remained the result of 250*250 pixels in the center of the tiled patch while discarding the edge with a length of 25 pixels.

We annotated the pixels in 23 cropped Landsat images into three land-cover types (ice, rock, and open water), serving as the ground truth objects. To enhance the classification capability of U-Net in various scales, the side lengths of Landsat images ranged from 30 pixels to 200 pixels. It is necessary to expand the sample set using data augmentation to prevent the network

from overfitting. Consequently, we augmented the annotated 23 sample images 20 times and obtained a total of 483 sample images. Our data augmentation methods include mirroring, translation, and rotation. Mirroring consists of three scenarios: vertical mirroring, horizontal mirroring, and vertical and horizontal mirroring. The translation involved a four-way translation up to 1/10 of the side length. The range of rotation angle was 0°-360°. Any void pixels that arose after data augmentation were filled by the reflecting adjacent image pixels. Among the 483 sample images, 80 % were randomly

assigned as the training set and the remaining 20 % as the validation set.

## 3.2 Open Water Identification

U-Net neural network is a deep learning network for semantic segmentation based on a fully convolutional network (Ronneberger et al., 2015), which requires less training datasets and time, compared with other neural networks for semantic

segmentation (Siddique et al., 2021). In addition, it does not require the explicit specification of the input image size for achieving end-to-end semantic segmentation. For LLOW detection, U-Net network can effectively fuse the spatial and spectral information. U-Net can process the spectral information for land-cover classification, while it can also consider the



spatial contexts to effectively reduce the interference of shadows and clouds. Thus, we implemented a U-Net model to detect open water in Landsat images and classify the pixels into three types of land cover: ice, open water, and rock. The

backscattering distributions of ice and rock are similar in single HH polarization, so we only classified the pixels of Sentinel-1 images into two types: open water and others. The single-channel input from Sentinel-1 images will lead to significant instability in the results of the U-Net model. To address this, we opted for a simpler approach for identifying open water in Sentinel-1 images. Instead of employing the U-Net model, we utilized a threshold segmentation method.

U-Net consists of an encoder and a decoder (Fig. 3). The encoder and decoder are both mainly composed of double-conv

layers, which contain double convolutional layers and are used to enhance model depth (Wu et al., 2020). In the double-conv layers, batch normalization layer and the Leaky Rectified Linear Unit (LeakyReLU) layer are added to re-correct the data distribution and achieve nonlinear computation. To avoid gradient disappearance and facilitate the deepening of the U-Net model network, we added a residual layer between the double-conv layers. In our U-Net model, the encoder expands the receptive field by using convolution layers and pooling layers to extract high-dimensional features. It is responsible for

encoding the seven Landsat bands into compressed features. These features then serve as the input for the decoder, which carries out the subsequent decoding and reconstruction of the spatial information. In order to improve the spatial granularity of decoding, the intermediate features with different spatial scales from the encoding part are also used as inputs for the decoder. These encoded intermediate features are concatenated with the decoded intermediate features through the skip connection layer. The transposed convolution layer between double-conv layers is used for the reconstruction of the spatial

details. The last convolution layer employs a convolution window of 1*1 pixels to classify the land cover through the decoded features.



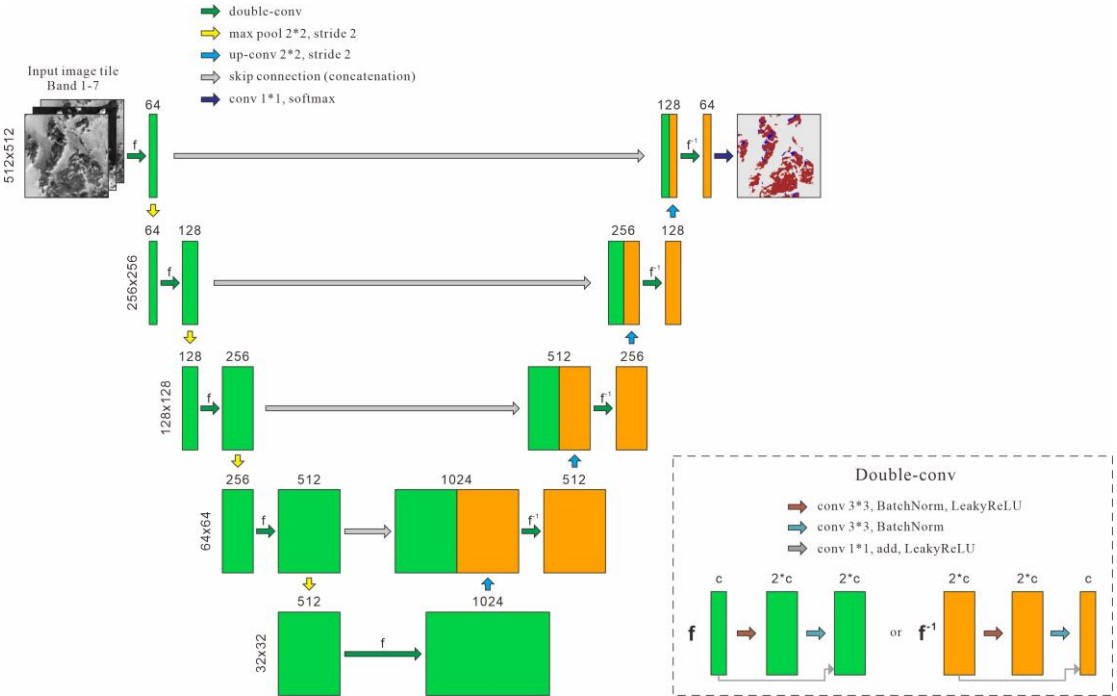

**Figure 3. The structure of U-Net and double-conv layer. The numbers on the left of double-conv blocks represent the image sizes. The number above the double-conv blocks represent the feature channels after each operation. Double-conv blocks are able to double the feature channels in encoder, while they halve the feature channels in decoder.**

We employed a threshold-based method to identify open water in SAR imagery. Typically, open water bodies exhibit a smooth surface, resulting in weaker backscatters, while areas with rougher surfaces generate stronger backscatters (Wang et al., 2019). Leveraging these backscatter characteristics, our threshold-based method classifies pixels with backscatter values below a specified threshold as open water, enabling effective identification of open water areas. The generation of thresholds involves both empirical and automated methods, relying on the grayscale histograms obtained from SAR images. The empirical method establishes a single global threshold applicable to all images, while the automated method generates various thresholds tailored to each individual image. Through a comparison of their performance, we found that thresholds derived from automated methods yield significant classification errors when applied to unimodal histograms, which aligns with previous studies (Yuan et al., 2015). During the months of January to April, when obvious freeze-thaw processes occur in landlocked lakes, the distribution of image histograms changes from unimodal to bimodal and then returns to unimodal. To mitigate classification errors in images with unimodal histograms, we used the empirical method to obtain the global threshold. Since the most prominent difference between the peak representing open water and the peak representing other areas in histograms often occurs in February, we examined some bimodal histograms from that month to find a suitable global threshold. The global threshold derived from February represents the general SAR characteristics of open water, so the global threshold can be applied to all Sentinel images in other months. We identified the most universal normalized



grayscale corresponding to the valley between these two peaks, namely -1.1, and determined this grayscale as the open water identification threshold for all Sentinel images.

### 3.3 Post-processing

Not all "open water" pixels extracted through the open water identification models are LLOW. Only open water areas which located within the rock area are regarded as LLOW. The identification of LLOW can be complicated by factors such as glacial rivers and melted water from coastal glaciers. Besides, LLOW may be indirectly surrounded by rocks. For example, LLOW may be enclosed by ice, which in turn is surrounded by rocks. In such cases, LLOW may directly contact the ice instead of the rocks but still be classified as being within the rock area. In our classified results, a classified Landsat image consists of a connected non-rock area and interspersed rock areas containing LLOW. The breadth-first search (BFS) algorithm has been proven to be effective in removing connected areas while preserving others (Silvela and Portillo, 2001). Thus, the BFS algorithm can effectively eliminate the connected non-rock area while retaining the rock areas. Subsequently, LLOW can be extracted from the remaining rock areas. In the BFS algorithm, we assume a four-connected adjacent relationship between pixels, meaning each pixel is directly connected to its four adjacent pixels in the up, down, left and right directions. According to the spatial characteristics of LLOW, LLOW pixels in gridded data are defined as those open water pixels, which should not be connected to any pixels in the image edges without traversing through any rock pixels. In other words, if an edge pixel is connected to some open water pixels without passing through any rock pixels, those open water pixels are not classified as LLOW pixels and should be removed. To perform the BFS algorithm, we initiate the search from all the edge pixels of the image. The search process proceeded by moving in the direction of the surrounding pixels that had not been searched or classified as "rock". During the search, the pixels that have already been searched are marked as "not LLOW". This search method simulates the spreading of seawater, continuously removing non-rock areas where contact with seawater could occur, and leaving only rock areas where stable LLOW may exist. Finally, all the remaining open water pixels derived from Landsat images are extracted and marked as "LLOW".

The usage of Landsat images in the visible and near-infrared bands is subject to severe cloud interference due to the cloud coverage along the Antarctic coast. As mentioned in Section 2.2, within four study areas and nine years, only a total of 79 Landsat images can be applied for LLOW detection. Therefore, the number of Landsat images with low cloud cover in the study area did not meet the requirements of our time series analysis. To improve the temporal resolution of LLOW time series, we used Sentinel-1 SAR images as supplements. SAR images are not affected by clouds but have limited spectral information and lack accuracy in distinguishing ground objects like rocks and ice. The open water identified solely from SAR imagery often includes substantial amounts of mountain shadows and lakes which were not surrounded by rocks. Without spatial information of rocks and ice, BFS algorithm is invalid to extract LLOW from open water. Consequently, SAR images only enable the identification of open water instead of LLOW. Using data from either Landsat 8-9 or Sentinel-1 alone cannot precisely capture the temporal variation of LLOW. However, combining the maximum lake extent derived



from Landsat with the results obtained from SAR images provides a better approach to achieve higher temporal resolution
and more accurate results (Miles et al., 2017). Thus, for each study area, we defined the pixels that are classified as LLOW
in multiple Landsat results as potential LLOW. Specifically, if a pixel was identified as LLOW two or more times across
different Landsat results, it was considered as a potential LLOW pixel. We aggregated these LLOW distribution images and
obtained the potential LLOW area in Landsat datasets. After that, we combined the Landsat and Sentinel images, using the

potential extents of LLOW and the open water derived from SAR, to generate the long-term series of LLOW.

Because previous cropped images had wide rectangular boundaries, they still retained large non-research areas. To ensure
consistency of the extracted LLOW in study areas, we delineated the more detailed coordinate boundaries according to the
irregular shapes of study area. Then, the detailed boundaries were used to narrow down potential LLOW regions. It is
important to note that identifying LLOW in SAR images can be challenging due to various factors, such as strong wind,

floating thin ice layers and sensor speckle noise (Dirscherl et al., 2021a). These factors can impact the backscatter of LLOW
and make accurate detection of LLOW difficult. For instance, congealed ice generates large bubbles, and the bubbles
entrained within ice layer enhance backscatters (Hirose et al., 2008). Consequently, LLOW covered by only a few floating
ice layers or affected by strong winds may exhibit higher backscatter coefficients and cannot be detected by our threshold
segmentation model. Instances of strong winds and floating ice have temporary effects on the entire study area and result in

significant underestimation of LLOW. Therefore, we disregarded those underestimated LLOW results and generated a
reliable long-term series of LLOW. The LLOW series combined Landsat and Sentinel images have a spatial resolution of 10
m and a time resolution of nearly 6 days.

### 3.4 Accuracy Assessment

To validate the performance of U-Net and LLOW identification, we manually annotated ground truth labels from Landsat 8-
9 OLI as the test datasets. For validation of U-Net model, several Landsat images were selected, and the pixels in the images
were annotated as "open water", "ice" and "rock" to serve as ground truth. However, directly annotating the ground truth of
LLOW based on SAR images is challenging and time-consuming, primarily due to their complex backscatter characteristics
(Huang et al., 2021). Therefore, we utilized optical image interpretation and subsequent transformation, in order to align the

ground truth from optical images with LLOW results, ensuring spatial consistency ((Liang and Liu, 2020). In our evaluation,
Landsat images were annotated into "open water" and "others" to serve as the ground truth of LLOW identification. To
maintain the spatial consistency between ground truth and projected results, the labeled Landsat images were resampled and
cropped to match the LLOW results. In specific, we focused on periods when ground objects can be more clearly
distinguished. A total of 17100 pixels were annotated for U-Net, and 225300 pixels were annotated for LLOW identification

to evaluate their performance.

The accuracy of classification models is estimated by confusion matrix and Cohen's Kappa (Dirscherl et al., 2020). The
formulas are presented in Eq. (1), (2) and (3).





$$\text{Kappa} = \frac{p_o - p_e}{1 - p_e} \tag{1}$$

$$p_o = \frac{TP + TN}{TS} \tag{2}$$

$$p_e = \frac{(TN + FP) * (TN + FN) + (FN + TP) * (FP + TP)}{TS * TS} \tag{3}$$

where TS is the total number of samples; TP is the number of true positive classified results; FP is the number of false
positive classified results; TN is the number of true negative classified results; and FN is the number of false negative
classified results in confusion matrix.

## 4. Results

### 4.1 Classification Results

Figure 4 illustrates the process and intermediate results involved the LLOW identification. The Landsat images were
accurately classified into open water, ice and rocks through the use of U-Net (Figs. 4e, 4f, 4g, 4h). Compared to the images
of false color band combination, the results derived from threshold segmentation contained large amounts of errors (Figs. 4m,
4n, 4o, 4p). For example, the smooth ice layers were misidentified as open water in Fig. 4p. However, we obtained the
information of potential LLOW areas through the results of U-Net. To rectify these errors, the mask of potential LLOW
areas were employed (Figs. 4q, 4r, 4s, 4t), significantly improving the accuracy of LLOW identification based on SAR
images.



**Figure 4. The intermediate images and results in the workflow of landlocked lakes' open water (LLOW) identification. The first row displays the Landsat images by the false color band combination 7-4-3 (RGB). The white regions in these images represent the void data in band 7, 4 or 3. The second row exhibits the classification results of U-Net. The third, fourth and fifth rows represent the Sentinel images, the results of threshold segmentation and the results of detected LLOW, respectively.**

Figure 5 shows the classification results obtained by U-Net for extracts from all Landsat test scenes. The U-Net network has generally shown good recognition performance across various terrains in all 4 study regions. In specific, it effectively mitigates the impacts of diverse brightness and contrast levels in VHs (Figs. 5a, 5b). Moreover, it accurately distinguishes mountain shadows from water bodies in LHs without any misclassification (Figs. 5c, 5d). Notably, in SO, the presence of ice undulations causes numerous shadows. U-Net correctly identifies these shadows as ice (Fig. 5e), which can be a challenging task when using threshold methods. In addition, in both the SO and CWM, there are partially melted lakes primarily



composed of ice, which appears grayish (Figs. 5e, 5f). U-Net successfully identifies these lakes as ice surfaces, preventing any overestimation of open water areas.

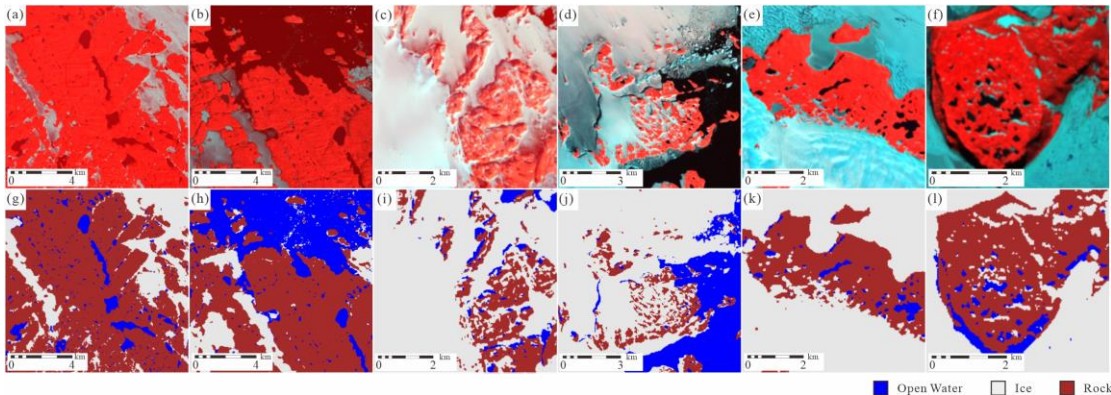

**Figure 5. Comparison between the Landsat images and auto-generated classification examples of U-Net. The upper row displays Landsat 8-9 images, using the false color band combination 7-4-3 (RGB), to enhance feature distinction. The lower row shows the corresponding auto-generated classification results of U-Net. Panels (a) and (b) represent Vestfold Hills (VHs); panels (c) and (d) represent Larsemann Hills (LHs); panel (e) represents Schirmacher Oasis (SO); and panel (f) represents Clearwater Mesa (CWM).**

Figure 6 displays LLOW results obtained through the fusion of Landsat and SAR images. A comparison within each row highlights differences between varied regions. For example, in April, the highest latitude region, SO, appears completely frozen (Fig. 6h), while lower latitude regions like CWM still exhibit LLOW during the same month (Fig. 6e). By contrasting the upper row with the lower row, temporal differences can be observed within the same region, where lakes show larger open water areas in the relatively warmer month of February (e.g., Figs. 6d, 6h)."






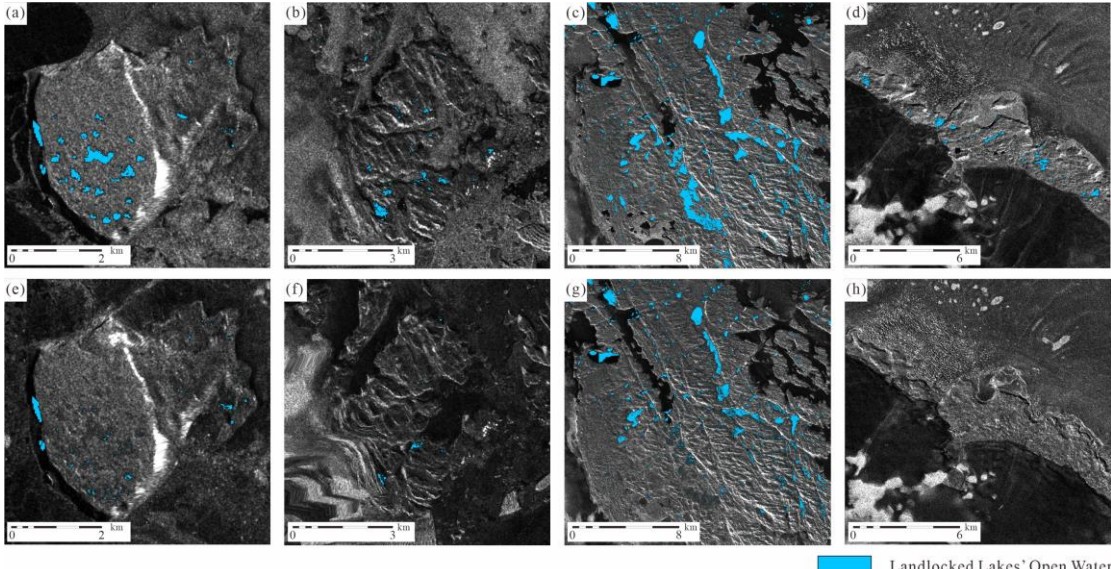

Landlocked Lakes' Open Water

**Figure 6. The landlocked lakes' open water (LLOW) area changes over time obtained through the fusion of Landsat and SAR images. The upper row shows the LLOW results in February, with lower row representing in April. Panels (a) and (e) represent Clearwater Mesa (CWM); panels (b) and (f) represent Larsemann Hills (LHs); panels (c) and (g) represent Vestfold Hills (VHs); and panels (d) and (h) represent Schirmacher Oasis (SO).**


## 4.2 Model Validation

We compared the accuracy of LLOW identification between results obtained before applying the potential LLOW areas mask and those obtained after applying the mask in LHs (Fig. 7). Prior to applying the mask, the threshold segmentation model identified a large number of false LLOW instances in low backscatter pixels. The LLOW identification only based on

SAR images resulted in a Kappa value of only 0.02 when compared to the ground truth labels. However, the masking process based on potential LLOW areas successfully reduced the majority of false LLOW instances and improved the Kappa value to 0.57. The increase in the Kappa value suggests that masking using potential LLOW areas can compensate for the lack of spectral information in Sentinel-1 images, thereby enhancing the accuracy of the LLOW identification model.



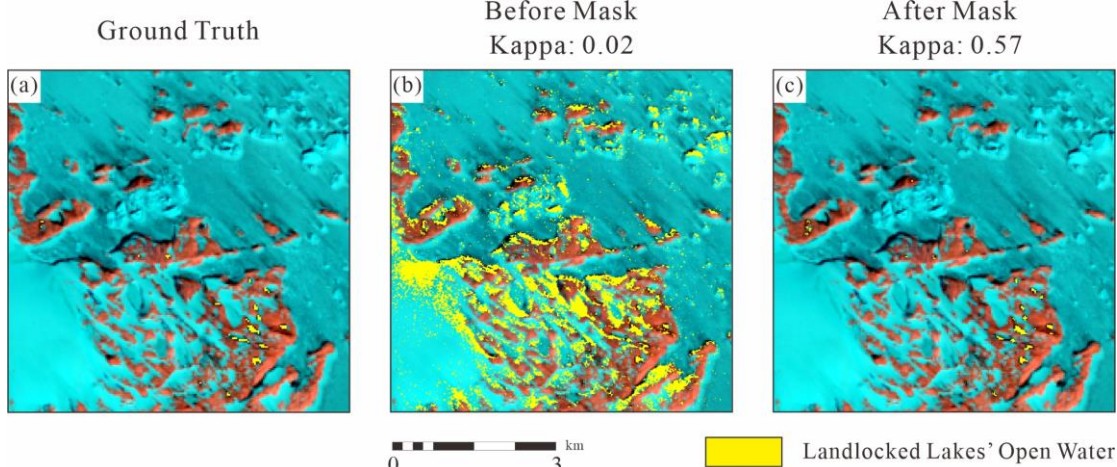


**Figure 7. Accuracy comparison of LLOW identification before and after mask in LHs. The background images are displayed from false color combination of 7-4-3 bands. The result before mask was derived from threshold segmentation.**

Our land-cover classification model, based on U-Net, achieved an average Kappa value of 0.85 on the test datasets,
indicating the reliable and accurate classification of land cover. The LLOW identification model yielded Kappa values ranging from 0.57 to 0.68, with an average Kappa value of 0.62 (Fig. 8). Among these, CWM exhibited the highest Kappa value of 0.68, suggesting the most similar spatial distribution between the predicted LLOW and the ground truth. On the other hand, LHs showed the lowest Kappa value of 0.57. For VHs and SO, the Kappa values were 0.59 and 0.64, respectively. In CWM, the locations and areas of LLOW were well recognized (Fig. 8i). In LHs, the spatial distribution of
LLOW was also accurately detected, although there were some inconsistencies in the boundaries between the ground truth and the predicted lakes (Fig. 8j). In addition, in VHs and SO, the model successfully identified large-scale LLOW areas exceeding 10,000 $m^2$, as well as a portion of small-scale LLOW areas below 10,000 $m^2$ (Table S2.). Overall, our model demonstrated proficiency in detecting larger LLOW areas and partially identifying small-scale LLOW areas, providing reliable information on the spatial distribution and extent of LLOW.



**Fig 8. Validation of landlocked lake identification model in testing dataset for four regions. The four columns of images are validation images for CWM, LHs, VHs and SO. The first, second and third rows are ground truth, predicted and spatial errors images, respectively. The background images are displayed from false color combination of 7-4-3 bands. The spatial distribution of classification errors is obtained from overlapping ground truth and predicted images.**

## 4.3 Seasonal variations in LLOW area

The study focused on changes in LLOW from January to April across four different regions in Antarctica. Figure 9 presents the spatial and temporal variations of LLOW area during the study period. Our results indicate an initial increase followed by



a decreasing trend in the overall LLOW area. Notably, the occurrence and duration of maximum LLOW areas varied among
the regions, with the highest value observed in early January in CWM, while the inland Antarctica region SO experienced its
peak LLOW area at the end of January, lasting for less than two satellite revisit cycles (12 days). The rate of decrease in
LLOW area slowed down from late March, approaching a relatively stable low-value stage. By April, the LLOW areas had
reduced to approximately 20 % of their maximum value for CWM, while the LHs and SO at higher latitudes decreased to 10 %
of their maximum or approached zero.

In addition to seasonal variations, interannual variations in LLOW area were observed. For example, LHs exhibited distinct
maximum LLOW areas in different years, with the maximum area recorded in 2021 being only 70 % of that observed in
2017. Furthermore, CWM experienced a significant freezing and thawing process in March 2017, where the LLOW area
dropped to less than 50 % of its maximum before subsequently rebounding to the maximum value.

**5. Discussion**

The changes in LLOW area can be categorized into two distinct phases: the growth phase and the decline phase (Fig. 9). The
growth phase spans from the initiation of our data collection until reaching the maximum LLOW area, while the decline
phase extends from the maximum area to the minimum area after reaching the peak. In the following sections, we will
discuss these two phases separately.






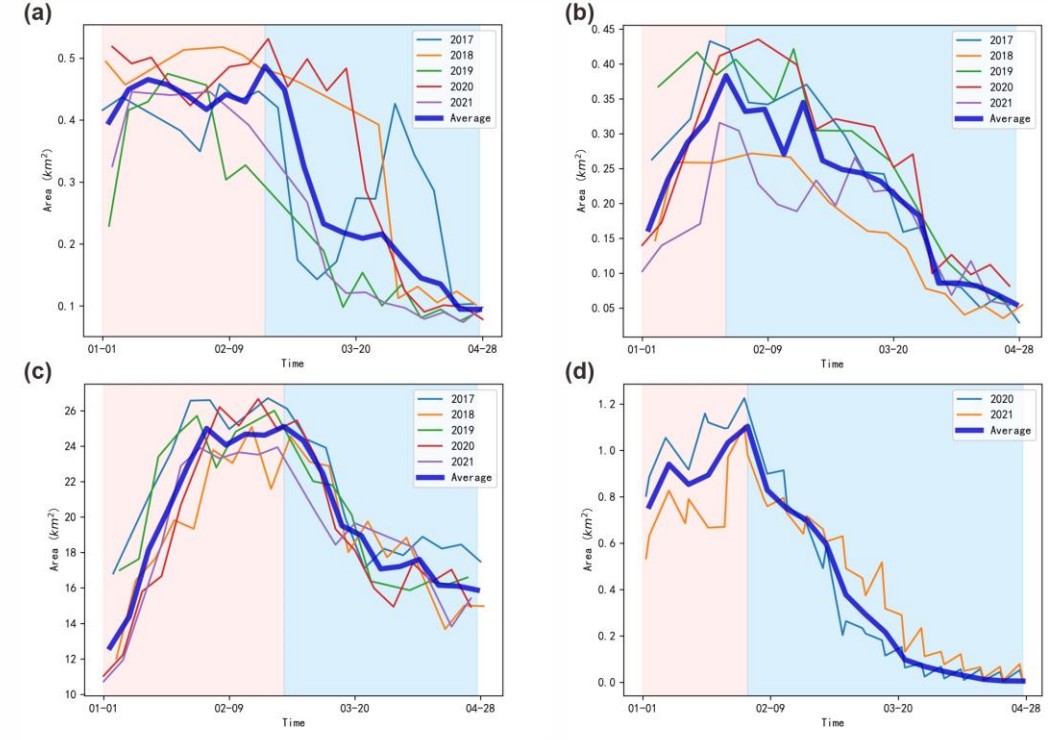

**Figure 9. The landlocked lakes' open water (LLOW) area changes in CWM (a), LHs (b), VHs (c), and SO (d) from January to April. The red interval represents the growth phase of LLOW area, while the blue interval represents the decline phase of LLOW area.**

**5.1 Growth phase of LLOW area**

With the onset of austral summer, lake surface ice and snow melt, resulting in the generation of meltwater, which contributes to an increase in the LLOW area. This process is closely associated with the changes in temperature, especially the occurrence of days with temperatures exceeding 0 °C (Braithwaite and Hughes, 2022; Li Qing, 2021; Wake and Marshall, 2015; Maisincho et al., 2014; Barrand et al., 2013). Thus, we evaluate the positive degree-day sum (PDD), which represents

the cumulative sum of temperatures above the melting point during a specific period (Cogley et al., 2010). In this study, the PDD for a given day is calculated as the sum of temperatures exceeding 0 °C from November 1st of the previous year until the current day. It is important to note that we only analyzed the PDD for LHs and VHs in this study, considering that the automatic weather station (AWS) data are only available in these two areas. Compared to the remote sensing-derived ERA5-land dataset, the AWS provides more precise measurements for calculating PDD. The PDD is calculated using the Eq. (4):

$$PDD_n = \sum_{i=0}^{n} \begin{cases} T_i, & T_i > 0 \\ 0, & T_i \leq 0 \end{cases} \tag{4}$$



Here, the positive degree-day sum prior to the day n is denoted as $PDD_n$ (°C) and $T_i$ represents the station mean temperature (°C) measured on day i. Fig. 10 illustrates the relationship between PDD and LLOW area change over time in LHs and VHs. During the growth phase of the LLOW area, the average $R^2$ value is around 0.9, indicating that PDD can explain ~90 % of the increase in LLOW area. However, there was a notable exception in LHs in 2018, characterized by an unusual cooling event from mid to late January. This event persisted for several consecutive days with temperatures below 0 °C, resulting in

a decline in the LLOW area. In addition, since the LLOW area had already reached its maximum at the beginning of January in LHs in 2018, the growth phase was short and less discernible, leading to a lack of significant correlation between PDD and the LLOW area.

PDDs can also influence the year-to-year fluctuations in LLOW area, but the relationship between changes in PDD and LLOW area is non-linear. For instance, the maximum PDD in 2017 was more than three times that in 2018 in LHs, while the

maximum LLOW area was less than two times that in 2018 (Figs. 10a and 10b). In VHs, the maximum PDD in 2018 was 50 % higher than that in 2017, yet the maximum area increased only by 10 % (Figs. 10f and 10g). Field observations indicate that lake ice first begins to melt near the shore due to greater heat transfer from adjacent ground surfaces (Peck et al., 2006; Michel, 1972; Watanabe et al., 1995). Therefore, ice in smaller lakes with more irregular shorelines may experience rapid melting even with lower PDDs, resulting in minimal changes in their LLOW area, even during years with significantly

higher PDDs. In addition, compared to LHs, the maximum LLOW area of lakes in VHs is less affected by PDD. This could be attributed to the fact that large water bodies in VHs are mostly epiglacial lakes or surface runoff, which were not considered within the scope of our study when extracting inland lake types. Thus, the overall LLOW area of VHs remains relatively stable under varying PDD conditions. These findings demonstrate that the LLOW area is not solely determined by PDD. Surface melt is expected to increase non-linearly with temperature because of feedback mechanisms such as the

snowmelt–albedo feedback and the wind–albedo interaction (Van Wessem et al., 2023). In addition, shoreline irregularity and lake area could also explain variations in ice-out events in certain lakes (Arp et al., 2013). Considering these factors, our results could have broader applicability.







**Figure 10. The positive degree-day sums (PDD) and landlocked lakes' open water (LLOW) area change during the 2017 (a), 2018 (b), 2019 (c), 2020 (d) and 2021 (e) melt seasons in Larsemann Hills (LHs) and during the 2017 (f), 2018 (g), 2019 (h), 2020 (i) and 2021 (j) in Vestfold Hills (VHs). In the figure, the red interval represents the growth phase of LLOW area The $R^2$ value in the figure is calculated from a linear fit of PDD and LLOW area during the growth phase.**

**5.2 Decline phase of LLOW area**

The freezing of lakes is closely related to the temperature drop (Kirillin et al., 2012). The relationship between the LLOW area and temperature in each region during the freezing season is significant (Table 1). The calculation of the $R^2$ value was based on a linear fit of the temperature and the LLOW area, ranging from the maximum LLOW area to the minimum temperature. The use of the minimum temperature instead of minimum area is justified due to the possibility of abnormal

temperature rebounds in March and April, which could lead to inaccurate fitting results between temperature and LLOW area. In all four study regions, the $R^2$ values were found to be greater than 0.5. This indicates a strong response of the LLOW area to temperature changes during the decline phase.

**Table 1. $R^2$ of the LLOW area and temperature in freezing seasons between 2017 and 2021.**

| Year | CWM | LHs | VHs | SO |
|---|---|---|---|---|
| 2017 | 0.74** | 0.92** | 0.59** | |
| 2018 | 0.63** | 0.96** | 0.28 | |
| 2019 | 0.73** | 0.83* | 0.88** | |
| 2020 | 0.86** | 0.64* | 0.95** | 0.7** |
| 2021 | 0.59** | 0.46* | 0.56** | 0.45** |
| Average | 0.71 | 0.76 | 0.65 | 0.58 |

* $p < 0.05$, ** $p < 0.01$.

During the freezing stage, the depth of a lake will affect the time of lake ice formation (Kirillin et al., 2012), thereby affecting the decrease in LLOW area. Shallower lakes tend to lose heat more rapidly, facilitating earlier ice cover formation. In contrast, deeper lakes possess greater heat capacity, resulting in a slower cooling process and delayed ice formation. For

example, considering the temperature difference between CWM and LHs, one would expect CWM to freeze and enter the phase of continuous decrease in LLOW area later. However, both regions exhibited a continuous decrease in LLOW areas starting from early February (Figs. 9a and 9c). This may be attributed to the range of lake depths in CWM, which spans from 0.2 to 4.6 meters (Lecomte et al., 2016), whereas LHs encompassed several lakes with depths exceeding 10 meters (Shevnina and Kourzeneva, 2017; Harris and Burton, 2010).



In CWM in 2017, abrupt declines and rebounds of the LLOW area were observed, returning to values close to the original

ones (Fig.11). When the temperature plummets, the LLOW area also drops rapidly and reaches its lowest point almost

simultaneously. When the temperature rises again, the LLOW area responds promptly, suggesting the dominant role of

temperature on the rapid changes in LLOW area.

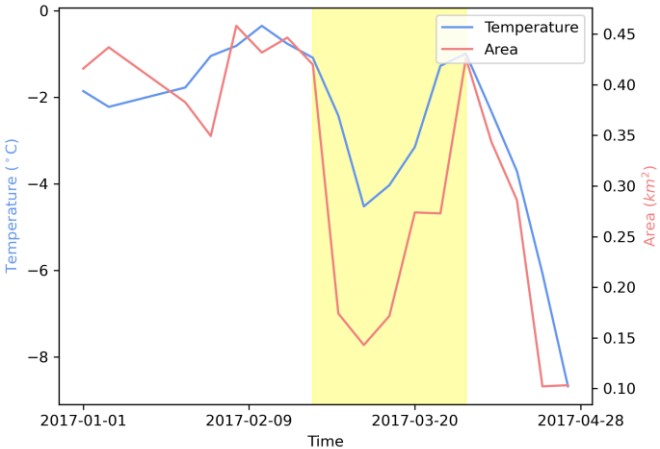

**Figure 11. The temperature and the landlocked lakes' open water (LLOW) area in CWM in 2017. The yellow interval represents the declines and rebounds of lake area and temperature.**

## 5.3 Model Limitation

The detection errors observed during the model validation can be attributed to two main reasons. Firstly, the unstable factors

such as strong winds and the presence of floating ice layers, led to the underestimation of the LLOW area. The backscatter of

LLOW can be influenced by the floating ice layer, strong wind, and snow covering open water, making accurate

identification challenging. By comparing our spatial error results with input SAR images, we found that the floating ice

layers directly caused large-scale lakes to be identified as fragmented lakes (Fig. 8l), while small-scale lakes were sometimes

overlooked (Fig 8k). Despite our efforts to remove significantly underestimated results, as mentioned in Section 3.3, these

unstable factors remain the primary causes of fluctuations in the LLOW area series. Furthermore, the presence of a blue ice

layer with low backscatters can lead to overestimation of LLOW. Secondly, the detection errors might have been

overestimated due to the limitations of our validation method. Specifically, the SAR images used for identification were at a

resolution of 10 m, while the Landsat images used for annotation were at a resolution of 30 m. The inconsistency in

resolutions resulted in discrepancies in delineating LLOW boundaries, leading to errors such as incorrect boundaries in LHs

(Fig. 8j). Consequently, this inconsistency introduced additional detection errors, including false positive errors and false

negative errors.

Although the LLOW identification model has these limitations, our findings demonstrate its strong performance across the four study areas. The performance assessment encompassed diverse environmental conditions such as various surrounding features, cloud covers, lighting conditions, and mountain shadows. This robustness suggests that our model can effectively detect LLOW in the entire Antarctic regions. By providing reliable long-term LLOW series products, our model contributes to a deeper understanding of the dynamic changes of LLOW under the influence of climate changes.

## 6. Conclusion

We proposed an automated detection workflow for LLOW based on deep learning and multi-source satellite images. By utilizing the BFS algorithm and combining Landsat 8-9 OLI and Sentinel-1 SAR images, we successfully distinguished the LLOW from other open waters, overcoming the limitation of models based solely on optical or SAR images. In our model accuracy assessment, our U-Net model and LLOW identification model achieved average Kappa values of 0.85 and 0.62, respectively, on the testing datasets. Our model accurately recognizes both large-scale and small-scale LLOW in the testing images. We selected four representative study areas in Antarctica: CWM, LHs, VHs, and SO. Applying our LLOW identification model to these regions, we mitigated cloud and shadow interference and generated high-resolution spatiotemporal LLOW area time series from January to April between 2017 and 2021.

The seasonal changes in LLOW area can be categorized into two phases: the growth phase and the decline phase. The growth phase includes the period from the initiation of our data collection until reaching the maximum LLOW area, while the decline phase extends from the maximum area to the minimum area after reaching the peak. We found that during expansion of LLOW area, ~90 % of the changes are explained by PDDs. PDDs can also influence the interannual variations in LLOW area, but the changes in PDD and LLOW area are not proportional. Furthermore, during the decline phase, air temperature changes accounted for more than 50 % of changes in LLOW area. Our model provides long-term LLOW series products that help us better understand how lakes change under a changing climate.

## Supplementary material

Please see the file of Supplementary material.

## Data Availability Statement

Data presented in this work are in the process of being hosted on a public server by the Chinese National Arctic and Antarctic Data Center (https://www.chinare.org.cn/).



**Author contribution**

GS conceived the study. XM designed the method and provided model data. AJ analyzed the data and interpreted the results.
AJ and XM designed and wrote the manuscript with the support of all co-authors. GS and YH improved the manuscript.

**Competing interest**

The authors declare no conflict of interest relevant to this study.

**Acknowledgements**

This work was supported by the National Science Foundation of China (Grant Nos. 42276243 and 41922046 to GS;
42071306 to YH), and the Program of Shanghai Academic/Technology Research Leader (Grant No. 20XD1421600 to GS).
The authors are grateful to CHINARE members for providing meteorological data.

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
