# Peer review of "Using Deep Learning and Multi-source Remote Sensing Images to Map Landlocked Lakes in Antarctica"

_EGUsphere, 2023_

## Author Comment (AC1)

***Summary***

*The manuscript aims to detect landlocked lakes in Antarctica fusing optical and SAR imagery and using a U-net based method.*

We would like to thank the Referee 2 for reviewing and commenting the manuscript. Below, we present a detailed response to each of your comments, with the original comments in italics and the responses in blue. All the recommended modifications will be implemented in the revised paper that will be uploaded.

***Novelty/Relevance***

*I'm not aware of another method addressing land-locked Antarctic lakes. However, the methods used are standard methods, or in the case of thresholding the SAR imagery outdated within the field of research. The thresholding method also means that the lakes under different wind states can't be separated. Moreover, the method can't separate these lakes from other types of lakes, not surprising, but if that was the goal the method needs to be further improved.*

We agree that the thresholding method is significantly affected by wind interference. The thresholding method didn't perform well with SAR images and separate landlocked lakes from other lakes, such as supraglacial lakes. But using potential landlocked lakes' open water (LLOW) areas derived from

Landsat images can extract landlocked lakes from other types of lakes in our workflow.

*Strengths*

*Study of land-locked lakes in Antarctica, where the method is designed to detect a specific type of lakes. Could this method be applied to other landlocked lakes to monitor water resources? If so then it might become a more viable method.*

Thanks for your suggestions for the potential application of this method. The landlocked lakes distributed in Antarctica are situated in diverse natural environments, such as cloud covers, terrain, mountain shadows and temperature variation. We trained the U-Net model to adapt these different conditions using various training data, such as thin clouds, mountain shadows and floating ice, which leads to the robust workflow for other study areas. What's more, the 6-day time interval of Sentinel-1 images contributes to the LLOW change monitoring. Thus, our method is able to monitor the variation of other LLOW areas in Antarctica.

*Weaknesses*

*The manuscript is weak in the technical details, in particular there is an apparent lack of understanding of satellite images and details around them are missing. How is the SAR data pre-processed? Is the different spatial resolution between the optical and SAR images accounted and corrected for? The incidence angle*

*dependency in SAR data will result in higher incidence angles having a lower backscatter response. How is this accounted for in the method? Are only repeat orbits used? How will the incidence angle affect the results here?*

Thanks for your suggestions for the pre-process of SAR images.

1) We used the Sentinel-1 level-1 GRD data as SAR images, which had undergone preprocessing steps such as radiometric calibration and thermal noise removal (https://sentinel.esa.int/web/sentinel/technical-guides/sentinel-1-sar/products-algorithms/level-1-algorithms/overview). We only conducted terrain correction on them. 2) The spatial resolution of Sentinel-1 images is 10 m. Landsat images maintained at a 30-m resolution until they are resampled to 10-m resolution after being processed into potential LLOW areas. 3) We didn't take measures to mitigate the effects of incidence angles. Considering the pre-process method proposed by Wangchuk et al. (2019), we will use the same relative orbits to reduce the influence of incidence angles. Fortunately, all SAR images we used are acquired in descending orbits except images in Schirmacher Oasis (SO). Thus, we will check the images in SO region. 4) The backscattering threshold between water and non-water are usually affected by the incidence angle (Wakabayashi et al., 2019). We will evaluate the influence in our results.

*Radar shadows (e.g. mentioned on P12 R271) are a well-known issue within SAR images. A method should be designed to deal with them or at least quantify the scale of the issue.*

A part of radar shadows has been excluded by the mask of potential LLOW areas during the identification of LLOW. We will evaluate the influence of radar shadows with DEM datasets in the revised version.

*Wind may cause a wind roughened (high backscatter) surface, it appears that the model can only deal with low backscatter surface scattering surfaces. The method would then only be applicable in a limited number of SAR images and this limitation would hinder an operationalization or a processing chain with a larger number of images. Moreover, separation of open water areas from surrounding is challenging due to the varying backscatter values under different wind conditions. To make a method applicable to be used in an ML/DL/operational setting all wind states need to be accounted for in the method. Something that is challenging for a threshold-based method.*

Thanks for your suggestion on the impact of wind. We agree that the threshold algorithm can't deal with complex wind condition and subsequent unstable backscatters. We had attempted to train deep learning models to identify water rather than threshold method. However, limited by technology and the size of training datasets, the deep learning model performed not well.

Thus, we used threshold method to extract water and focused on the identification of LLOW.

*The text should be significantly shortened to avoid unnecessary repetitions, focus the message on what was done here (without repetitions). E.g. among other things can section 3.4 be significantly shortened by removing repetitions. As the other reviewer has already pointed out that the text is verbose and provided examples I'll not do so further here.*

Thanks for your suggestion. We will make the article more concise and avoid repetitions.

*Stating that "best" analysis etc has been used should be strengthened to indicate what makes this the "best".*

Thanks for your suggestion. The U-Net model shows good performance across various terrains and conditions. For example, it is able to overcome the interference from the clouds, shadows and floating ice. The robust U-Net model contributes to the generation of potential LLOW areas for the entire Antarctica. The thresholding method also performs similarly across all regions. Thus, we proposed that our workflow for LLOW identification is robust and can be applied to the entire Antarctic area. This point will be clarified in the revised manuscript.

*The correlation between PDD and lake area is long established and is not new, and neither is different lake shape evolution with different PDD evolution. The fact that lakes melt first from the edges is fundamental knowledge and not new knowledge established here. Combined can these results easily be referred to in existing literature, and this manuscript should highlight what is new knowledge aside from these well-established results.*

We appreciate your comments. We acknowledge that these concepts are indeed well-established within the scientific community and are not presented as novel findings within our manuscript. In the original version of the manuscript, we simply provided a comprehensive background. We will remove these points in the revised manuscript.

*Figure 9 show significant amount of lake area before the start of the study, in order to show time series of lake evolution at least for 3 of the sites the time series needs to be expanded to include data from at least one month earlier. How is the time series affected by removing all the troublesome images, e.g. the wind affected and those where the method failed? Lack of useful data at the start and end of the season will lead to under/over estimation of the lake area. Can the method (time series) be said to satisfactory deal with rapid changes? Or is there a need to increase sampling frequency? P20R402. How does the lack of data in December affect this? It appears that for at least the CWM side data from earlier months is*

*needed to establish maximum lake area and probably also the LH site judging from Figure 9.*

1) Affected by strong wind and floating ice, there are often large areas of backscatter increase in lakes. The backscatter of the entire lake can rise to a level similar to the surrounding rocks or ice, making them indistinguishable. This phenomenon can lead to a decrease in the identified LLOW area by 10% to 50%. Therefore, when we remove these images, the time series become much more stable. 2) We agree that lack of valid data at the start and end of the season will lead to little understanding about minimum LLOW areas in frozen state. However, due to the 6-day interval between consecutive Sentinel-1 images, the LLOW time series can capture the maximum of LLOW areas. Thus, the current sampling frequency is sufficient. 3) The sustained high LLOW areas in the CWM from January to February indicate that the area in January is already the maximum extent of the lakes. The lakes won't freeze in January, suggesting that the area in December is expected to be smaller than that in January. Therefore, the absence of data for December does not lead to an underestimation of the maximum lake area.

*There are 3 different lakes presented here, how can this method separate the three types if they all exist in one satellite image?*

These three types of lakes, supraglacial lakes, epiglacial lakes, and landlocked lakes, have distinct characteristics. Supraglacial lakes are surrounded by ice

rather than the rock. Epiglacial lakes have only a partial contact with rocks but is not within a rocky area. Landlocked lakes are entirely situated within a rocky area. After land cover classification with the U-Net model, BFS can distinguish their relative positions to rocks in one image. The positional information of LLOW obtained from Landsat can be used as a reference for identifying the location of LLOW in Sentinel-1 images. Finally, landlocked lakes can be distinguished from other types of lakes in Sentinel-1 images.

*P13R290-292. If the LLOW are underestimated who does this affect the biological component that has been used as an argument for conducing the entire study?*

We appreciate your attention to this. If LLOW are underestimated, it could lead to a conservative assessment of available habitats for various aquatic species. This, in turn, could affect our understanding of biodiversity, ecological interactions, and the potential for conservation efforts within these ecosystems. Such underestimation might also impact the accuracy of ecological modeling and predictions regarding the distribution and abundance of species, which are crucial for formulating effective conservation strategies and understanding ecosystem dynamics in coastal Antarctica.

*P14R317. It is stated that using the thresholding method produces large amount of errors. Establishing an improved method should therefore be a goal of this manuscript. Open water areas that are either sea water, melt lakes on the ice*

*sheet or lakes on land is not possible using simple thresholding in the SAR images as the radar signal interprets each as water.*

Thanks for your suggestion. The interpretation of SAR imagery is challenging due to ambiguous backscatter returns and image geometry effects (Li et al., 2021). Due to technological limitations and the scale of training datasets, we were unable to implement the deep learning for water identification with SAR images. Thus, we selected the thresholding method to replace it. In addition, the utilization of the potential LLOW areas also aids in the thresholding method to eliminate the interference from shadows and other types of water like supraglacial lakes.

*P21R411-420. Lake growth after temperatures start exceeding 0 has been well studied on the Greenland ice sheet for well over 10 years now. And PDD was used by, e.g. Johansson et al, (2013) to study lake evolution.*

*Johansson, Jansson and Brown (2013): Spatial and temporal variations in lakes on the Greenland Ice Sheet, J. Hydrology,*

*https://doi.org/10.1016/j.jhydrol.2012.10.045*

We greatly appreciate the reviewer's insightful comment and the reference to the seminal work of Johansson, Jansson, and Brown (2013) on the spatial and temporal variations in lakes on the Greenland Ice Sheet. We acknowledge the importance of this prior research and its relevance to our study, which

explores the growth of lakes' open water area in response to exceeding zero-degree temperatures, utilizing Positive Degree Days (PDD) to assess lake evolution. We will include this work in our reference list.

***Presentation***

*There is a substantial amount of details about chemical and biological importance of these lakes in the introduction. Shorten this to one paragraph, up to 4 sentences and highlight instead how this work fits into lake detection using satellite images, ML/DL of lake detection, or similar. The work should be set into the context of existing science with the topic presented here not in a different scientific field.*

Thanks for your suggestions about the introduction. We will condense the mentioned details into one concise paragraph, comprising up to four sentences. This revision will emphasize how our work integrates with and contributes to the existing body of research on satellite-based lake detection and the application of ML/DL methodologies in this field.

*P7R145-150. If dual-polarization data is not available, why is it being discussed here where the method is presented? This would then fit better in the introduction or the discussion.*

Thanks for your suggestions. We will move this section to introduction.

**Minor comments**

*P2R12 what is "reliable" in this context?*

We appreciate the reviewer's question concerning the use of the term "reliable" in the context of Antarctic landlocked lakes' open water (LLOW) serving as a climate indicator. In our manuscript, the term "reliable" is intended to convey the consistent and predictable nature of LLOW as an indicator of climatic conditions in the Antarctic ecosystem. LLOW's sensitivity to climatic variables, such as temperature and precipitation, enables it to indicate the broader climate information. This sensitivity is based on the direct relationship between climatic factors and the physical, chemical, and biological processes occurring within these lakes, making them effective indicators of climate change.

*P2R16. Why did you choose ice-free areas? And what do you mean with ice-free here, no glacier ice, no inland ice sheet, no sea ice.*

Thank you for your question. The ice-free areas refer to the coastal Antarctic continental areas without ice or snow during the austral summer; these are regions without glacier ice and inland ice sheets. In these ice-free areas, lakes undergo freezing and melting cycles, playing a crucial role in maintaining the ecosystems of Antarctica. Moreover, changes in the areas of these lakes may be sensitive to climate warming. Thus, a thorough understanding of the areas

of these lakes is of great significance for assessing the impact of climate change on Antarctic ecosystems. However, current methods for detecting this kind of lakes are unavailable. Thus, we aim to develop new techniques and methodologies to improve the detection and analysis of this type of lakes, contributing to the broader field of climate and ecological research in Antarctica.

*P4R63. Rapidly -> change to a more scientific wording.*

Thanks for your suggestion. We will correct.

*Section 2.2. The number of optical images are introduced as 79 and then there is the discussion about removing image. It later transpires that 79 images were used. The text must be amended so that it is clear how many images were being used.*

Thanks for your comment. We used 79 Landsat images and 390 Sentinel-1 images at the beginning. Then in post-processing section, we remove 45 Sentinel-1 images which are affected by strong wind or other factors. Thus, we used a total of 345 Sentienl-1 images. We will correct the text in the revised paper.

*P5R103. Specify what "superior in many aspects" means.*

Thanks for your suggestion. We will add the explanation about it.

*P6R113. This is well known remove this reference to fundamental radar knowledge.*

Thanks for your suggestion. We will remove this reference.

*P6R117. Define high-resolution*

Thanks for your suggestion. We will clarify the temporal resolution.

*P6R125-127. Why is there a reference attached to one of the weather stations and not the other? Is it possible to give credit to the data providers instead? The text about "temperature" is confusing, is this not actual temperatures but some kind of simulated temperatures or why has been used?*

Thank you for your comments. Regarding the Davis station data, it is from the Australian Government Bureau of Meteorology's official website at http://www.bom.gov.au/climate/data/stations/, with the station number being 300000.

As for the term "temperature", it refers to "daily mean air temperature" and "daily mean near-surface temperature". The quotation marks here emphasize this abbreviated expression. Please be assured that the station data represents actual observations. It is not derived from reanalyzed data sources like ERA-5.

*Chapter 3. The ground truth should be presented in the data and not as a part of the results in chapter 3. This also goes for parts of chapter 3.2 that should also be moved to the data section.*

Thanks for your suggestion. We will move description of ground truth to chapter 2. Chapter 3.2 is about the process of open water identification, so it's not suitable for the data section.

*P11R226-229. Very well known (fundamental radar) remove reference.*

Thanks for your suggestion. We will remove this reference.

*Within this manuscript Sentinel-1 has been used, this is essential to call it or make an acronym if it's preferred to call it Sentinel. This as there are many ESA Sentinel satellites, and there is also the Asian Sentinels.*

Thanks for your suggestion. We will uniformly replace "Sentinel" with "Sentinel-1".

*P12R270. Many methods detect glacier outlines etc. A more thorough method should be able to at least attempt to separate ice (moving materiel) from the more stationary rocks.*

Thanks for your suggestion. We will revise this incorrect description.

*P13R297-299. Well known fundamental radar knowledge, remove reference.*

Thanks for your suggestion. We will remove this reference.

*P16R348. Remove "*

Thanks for your suggestion. We will remove it.

Reference

Li, W., Lhermitte, S., and López-Dekker, P.: The potential of synthetic aperture radar interferometry for assessing meltwater lake dynamics on Antarctic ice shelves, The Cryosphere, 15, 5309-5322,https://doi.org/10.5194/tc-15-5309-2021, 2021.

Wakabayashi, H., Motohashi, K., and Maezawa, N.: Monitoring lake ice in Northern Alaska with backscattering and interferometric approaches using Sentinel-1 Sar Data, IGARSS 2019-2019 IEEE International Geoscience and Remote Sensing Symposium, 4202-4205,

Wangchuk, S., Bolch, T., and Zawadzki, J.: Towards automated mapping and monitoring of potentially dangerous glacial lakes in Bhutan Himalaya using Sentinel-1 Synthetic Aperture Radar data, Int. J. Remote Sens., 40,

4642-4667,https://doi.org/10.1080/01431161.2019.1569789, 2019.

---

## Author Comment (AC2)

*Summary*

*The paper describes a semantic segmentation scheme to map landlocked lakes in Antarctica, using Landsat and Sentinel-1 satellite imagery as base data. Landsat images are segmented with a U-net, Sentinel-1 with a manually tuned threshold. The results are merged with a simple late fusion logic.*

We would like to thank you for reviewing and commenting the manuscript. Below, we present a detailed response to each of your comments, with the original comments in italics and the responses in blue. All the recommended modifications will be implemented in the revised paper that will be uploaded.

**Novelty/Relevance**

*There isn't any technical novelty. Methods are standard and used in somewhat ad-hoc manner without clear justification for the design.*

We agree that our methods are standard which had been widely applied, such as the application of the U-Net model to detect supraglacial lakes in SAR images (Dirscherl et al., 2021b). We also used U-Net model for landlocked lake detection with Landsat images. However, surrounding information such as rock and ice is necessary for the identification of landlocked lakes, while this information is unavailable in previous studies. Thus, we attempted to identify the surrounding situation of lakes through multi-classification and

breadth-first search (BFS). We generated potential landlocked lakes' open water (LLOW) areas as a water mask, and successfully transferred surrounding situations of lakes from Landsat images to Sentinel-1 images through the mask process.

*The specific application appears to be new, I am not aware of any paper that described the specific case of land-locked Antarctic lakes. That being said, the distinction is perhaps a tad contrived, there has certainly been work on detecting supra-glacial lakes, so the only difference is really to check whether a lake is surrounded by rock or by ice.*

We agree that the primary difference between our work and previous studies is the detection of whether a lake is surrounded by rock or by ice. But the work is challenging and time-consuming. Limited by the complexity of rock and ice backscatters, SAR images can't distinguish rock and ice. Due to the interference of clouds, multispectral images are not suitable for generating long time-series datasets. Only the combination of SAR images and multispectral images can solve this problem.

**Strengths**

*Since the task has apparently not been studied before, there is potential to systematically map land-locked lakes with the method (it is not done at any scale,*

*though). I am not an expert in Antarctic ecology or climate and cannot judge the relevance of this, but it is a mapping capability that hadn't been investigated.*

*The proposed method works moderately well, even if the segmentation performance is not surprising or spectacular for the fairly straightforward task.*

We would like to thank you for the positive comments on the significance of this work. In fact, the threshold method didn't perform well on the lake detection task. However, the detection of water in the Antarctica is more difficult than in other areas. Interfered by freeze-thaw process, strong wind and incidence angle (Dirscherl et al., 2021a; Li et al., 2021), the backscatters of water become unstable, which usually change significantly.

***Weaknesses***

*Technical decisions seem somewhat arbitrary and ad-hoc. Not seriously wrong, but the described scheme is just "a way to do it", not a carefully designed and justified "best way to do it".*

Thanks for your suggestion on our workflow. After our exploration and experiment, each step in the workflow is necessary for completing this task. For example, we employed the upsampling method and adjusted the patch size, to better identify tiny landlocked lakes with a few pixels in images. When the number of valid Landsat images is insufficient to form a LLOW time series, we changed the results derived from Landsat images into masks. Then we

combined them with Sentinel-1 images to improve temporal resolutions of the LLOW time series. In addition, we also changed the parameters in the U-Net model for the better model performance.

*The evaluation is rather weak, using only a few small areas, and even excluding some lakes that are clearly visible within the image tiles. The study does not go beyond the four small proof-of-concept regions, there are no large-scale, wall-to-wall results.*

We agree that some obvious lakes were excluded in our results. When identifying potential LLOW areas, these lakes are not surrounded by rocks in classification results. We will attempt to improve our model performance to avoid the misidentification. In figures 4,5, and 6, we only present parts of the results. Considering that some LLOW areas are relatively small, especially in LHs, we increased the map scale and didn't show the entire area. We will add new figures to display the wall-to-wall results in the revised manuscript.

*The model validation suggests that almost all the performance is due to Landsat, whereas Sentinel-1 does not offer much except the potential to densify in time - which however is not actually done, since the Landsat segmentation acts as a hard constraint: the algorithm does not appear to allow SAR to add lake pixels.*

We agree that the potential LLOW areas strictly constrain the potential of SAR images in the current algorithm. We will draw a buffer around the potential

LLOW areas (Wangchuk et al., 2019) to allow water to extend beyond previous masks.

*Finding of a "decreasing trend in LLOW area" for 2017-2021 is rather trivial and expected. It would be more interesting to interpret the measured areas beyond just that obvious trend.*

We thank you for the insightful comment. The reliance on available satellite imagery inherently limits the scope of our trend analysis. Despite our efforts to utilize the most comprehensive and up-to-date datasets, the frequency and resolution of available images restricted our ability to identify and interpret more complex trends in LLOW dynamics throughout the study period. In particular, the temporal resolution of the imagery posed challenges in capturing short-term fluctuations or subtle changes in the LLOW area, which could potentially reveal more about the underlying processes affecting these water bodies.

**Presentation**

*Throughout, the text could be made shorter and more concise. E.g.,*

- *lines 215-220 are unnecessary, everything that is said there is already implied by the use of U-net*

We will delete this section of text.

- *228-240 is a verbose, meandering way to simply say "we manually chose a global threshold by inspecting histograms"*

We will shorten the description of threshold selection process.

- *250-264 says little more than that the definition of a land-locked lake is a water region surrounded by a rock region.*

- *etc.*

In this paragraph, we describe the process of BFS in detail and explain how a land-locked lake is identified. We will shorten this paragraph and simplify other wordy sections of the paper. Thanks for your suggestion.

*The introduction is verbose and not very focussed, touching on all sorts of studies about land-locked lakes that have no relation or importance for what the paper then does.*

Thank you very much for your comment on the introduction part. We will shorten the introduction, and focus on the lake identification.

*The analysis in lines 420-440 is rather hand-wavy, I was not able to see what purpose it actually serves. It gives me the impression that the authors just performed a random analysis that was easily doable, to send a message that the maps could potentially serve some useful purpose.*

We appreciate your comments. The purpose of this section was to elucidate the relationship between Positive Degree Days (PDD) and the area change of landlocked lake open water (LLOW) in different areas. We recognize the reviewer's concern that the analysis may appear cursory or lacking in clear purpose. To address this, we offer the following clarifications and enhancements to underscore the relevance and rigor of our analysis:

The primary purpose of the analysis was to demonstrate the influence of PDD on changes in LLOW areas with time. By showing a strong correlation (average $R^2$ value of approximately 0.9) during the growth phase of LLOW areas, we aimed to validate PDD as a key factor explaining most of the variability in LLOW area.

We detailed an exception in 2018 for LHs during an unusual cooling event to show that while PDD is a strong indicator of LLOW area changes, other climatic and geographic factors can also influence the outcomes. This exception was included to prevent oversimplification and to encourage a nuanced understanding of LLOW area dynamics.

*There are remaining language issues, both in terms of English grammar (random example: "due to non-uniform of field surveys") and in terms of technical expressions (e.g., "gradient disappearance" instead of "gradient vanishing").*

We appreciate the reviewer's comments.

We will thoroughly check the whole manuscript. Also, we will revise the points raised by you, following your suggestions.

***Technical Questions***

*The computational procedure is not entirely clear. It is one way of cobbling together a segmentation pipeline, but there is no clear explanation why that specific design was chosen. Whereas there are obvious concerns about it, e.g.,*

- *the potential benefit of Sentinel-1 for the land-cover map are not exploited*

The mask process between Sentinel-1 and land-cover map can be modified using the buffer method mentioned above. We will revise the workflow and results in the revised manuscript.

- *possible correlations between optical and SAR are lost*

Ensuring that optical and SAR image dates are close to each other is challenging in identification of LLOW in Antarctica. Thus, we aggregate all Landsat images into one potential LLOW area instead of the fusion of each Landsat and Sentinel-1 image.

- *the fusion seems to not leverage away the (pseudo-)probabilistic segmentation scores*

Our mask method follows the priority principle for potential LLOW areas identification. Any water pixel not within the potential LLOW areas is not

LLOW. We didn't consider probabilistic scores in the fusion process. The combination of buffered masks and probabilistic scores might improve our model and allow SAR images to add pixels. We will try this method and test the performance in the revised version.

*What is meant by saying 300x300 is the "common" patch size for U-net models? There isn't a single, canonical patch size for training those models, and at test time they are anyways fully convolutional and not tied to a specific patch size.*

This is an inaccurate description. We will correct it. We had used this parameter and the word "common" because we referred to previous publications and code. In addition, we agree that fully convolutional networks can deal with patches with various sizes. However, when we standardized the size of training patches, it did perform relatively well compared with the method without resampling.

*I don't understand the upsampling of the input for the U-net. No information is added by this and the effective receptive field / context window inside the network is actually reduced. So it would seem that one can reach at least equal performance, with lower computational effort, by properly training the U-net to handle the smaller images. Please explain.*

The 1024*1024 patches consume more computational resources than 300*300 patches during the U-Net inference. However, as described in the

paper, the upsampling is used to magnify the small open water area. Some open water areas only consist of a few pixels in the images. Magnifying these areas through upsampling contributed to the identification of small LLOW.

*Certain augmentations should be ablated and empirically justified. Conceptually it seems problematic to apply transformations like rotating or vertical flipping, as this leads to illumination directions that are implausible in real Landsat images, especially at Antarctic altitudes with low sun elevation.*

Thanks for your suggestion. We didn't consider the impact of image augmentations on the direction of illumination and shadows. Flipping and rotation disrupt the regular shadow features. These shadow features might benefit the training process. We will try a new augmentation strategy and compare the model performance.

*Using a threshold for segmentation is of course entirely correct and sensible, if it works. But the justification that single-channel input will lead to "instability" of U-net makes no sense. Countless applications use U-nets with various single-channel inputs (SAR, panchromatic, depth,...).*

Thanks for this comment. This is an inaccurate description. We will correct it. We agree that many studies had used U-Net models with single-channel inputs. But in our process of training U-Net model for SAR images, the U-Net

produced several instable results. For example, in snow-cover areas, the snow with relatively low backscatters were identified as water by mistake.

*The fusion step is unclear. First you argue for using SAR, and for combining it with optical data, to obtain better temporal resolution. But then, a consensus over at least 2 Landsat acquisitions is required for a potential LLOW pixel, meaning that the shortest possible resolution of everything that follows is the interval between three Landsat overpasses (if a pixel flips from ice to water between two consecutive images, you need to wait for a third image to confirm, so over the entire period you cannot say whether the pixel remained the same or thawed and froze again).*

We used all land-cover maps during five years to generate one potential LLOW area map for one region. Among several dozen images, if a pixel is identified as LLOW in at least two images, then that pixel is considered a potential LLOW. This algorithm aims to preserve as much potential space as possible.

*In line 290, it remains unclear how the authors "disregard" underestimated lakes. To do that one must identify them first - but the algorithm, by definition, does not know where it made an underestimation error.*

We manually deleted those images which underestimated the LLOW. In the melting phase, the backscatters of water may suddenly increase due to

strong wind or other reasons (Fig. 1). We will check the images where the lake areas suddenly decrease in the time series curve. If backscatters of water increase abnormally, we will manually remove these images.

[Figure]

Figure 1. The significant backscattering changes of lakes during 25 January, 31 January and 6 February in the VHs.

*I do not understand why only a tiny set of 17k pixels were "annotated for U-net", but 225k pixels were "annotated for LLOW identification". 17k seems an overly small training set: assuming an average lake size of, say, 600x600m that would be fewer than 50 lakes. Why would one do that if apparently another 225k annotated pixels are available?*

The labels annotated for U-Net and LLOW identification are different. The labels for U-Net consist of water, rock and ice, while the labels for LLOW only consist of LLOW and non-LLOW. Thus, we can't apply LLOW data to U-Net testing.

*Cohen's kappa as a segmentation metric is discouraged (cf. [Pontius and Milliones, 2011]). I would recommend to follow best practice and show confusion matrices, F1 scores, IoU scores.*

Thanks for your suggestion. We will add new accuracy validation experiments on F1 scores and IoU scores.

*The PDD metric (equation 4) is defined in a strange manner. According to the definition that metric is exactly the same for a cold spell with two weeks of constantly at 0 degrees, or of constantly -25 degrees. Surely that would make a difference for the ice cover of the lakes? Wouldn't it be more natural to look at the number of consecutive PDD, or to integrate the average temperature including negative values?*

Thank you for the insightful comments and suggestions. The use of the PDD metric in our study is grounded in its common application within glaciological and climatological research as a simplified measure to estimate the melting potential of ice and snow. The rationale for focusing exclusively on

temperatures above 0°C is to directly quantify the thermal energy available for melting ice, which is a critical factor in the dynamics of lake ice cover.

However, you raised an important point about the continuity of temperature conditions during the melting phase. It is essential to highlight that temperature patterns preceding the accumulation of positive degree-days (PDDs) typically show a gradual increase towards 0°C. This pattern means that, even before reaching the threshold for PDD calculation, the thermal conditions are already near the melting point. Thus, a cold spell with two weeks of constantly at -25 degrees during the melting phase would be unusual.

To assess the influence of higher temperatures more accurately on the expansion of open water areas, we utilized the average daily temperature, inclusive of negative values, to calculate PDDs. This method acknowledged that any day with an average temperature above 0°C contributes to the melting process, providing a nuanced measure of thermal energy input relevant to ice melt. However, days with an average temperature below 0°C were considered to have a negligible impact on ice melting and were thus not included in our PDD calculations.

*For the decline (Section 5.2), why use the minimum temperature? To my knowledge, and also more in line with the PDD metric used earlier, a more*

*common indicator is the number of consecutive negative degree days, at least in studies of lake ice in Canada, the Alps, etc.*

We appreciate your comment. In our study, we opted to use the average temperature rather than the minimum temperature for analyzing the decline phase of lake ice cover. The mention of minimum temperature in our text relates specifically to the context of defining the range for our correlation analysis between LLOW area and temperature metrics. This was articulated in the manuscript to delineate the bounds for calculating the $R^2$ value based on a linear fit, aiming to refine the focus of our correlation analysis.

We acknowledge the common practice of using consecutive negative degree days (NDDs) in lake ice studies across regions such as Canada and the Alps. This practice serves as a direct measure of the cold spell's duration and intensity, which influences ice formation and maintenance.

The choice to exclude NDDs as the main measure in this stage of our research is based on previous analysis results, which showed that although NDDs play a significant role in initiating ice cover formation, their connection to the shrinking size of LLOW area does not follow a linear pattern (Graf and Tomczyk, 2018). Our observations suggest that, once the lake ice cover is developed, its dynamics are more closely correlated with average temperature variations.

*Minor Comments*

*Why do ice-covered lakes "magnify the warming trend"? I would think they might rather dampen it?*

We appreciate the reviewer. Ice and snow typically have high albedo, reflecting a significant portion of incoming solar radiation, thus potentially having a cooling effect on the local environment. However, the context of our discussion focuses on the transition period from ice-covered to ice-free conditions and the consequent alterations in surface albedo and energy absorption.

The mechanism by which ice-covered lakes can amplify warming trends is linked to the concept of the ice-albedo feedback. During the winter and early spring, ice-covered lakes indeed reflect a large portion of solar radiation due to their high albedo. However, as temperatures rise, ice begins to melt, reducing the surface area covered by ice and exposing the underlying water. Liquid water has a significantly lower albedo than ice, indicating that the lake absorbs more solar radiation instead of reflecting it. This increased absorption of solar energy by the water surface not only contributes to further warming and melting of the remaining ice but also leads to a rise in water temperature, which can enhance local warming. This mechanism aligns with observations and modeling studies that have documented the significant role of ice-albedo feedback in accelerating warming, particularly in polar and

high-latitude regions where ice and snow cover are integral to the climate system.

*It is a strange claim that U-net "requires less training datasets and time" than other neural networks. That depends on who you compare to, of course there are designs that are faster than U-net (e.g., those created for mobile or embedded devices). Moreover, "U-net" isn't a specific architecture but a whole family of networks with certain characteristics - essentially, symmetric hourglass encoder-decoder structure with dense skip connections. So some "U-nets" are a lot slower and more data-hungry than others.*

Thanks for your suggestion. We will revise this description following your comments. The U-Net model was originally designed to work with fewer training datasets for biomedical image segmentation, but the landscape has evolved considerably since then. Many new architectures have emerged, designed for accelerated training and optimized for smaller scales. In addition, parameters of U-Net also determine the computational cost.

Reference

Dirscherl, M., Dietz, A. J., Kneisel, C., and Kuenzer, C.: A Novel Method for Automated Supraglacial Lake Mapping in Antarctica Using Sentinel-1 SAR Imagery and Deep Learning, Remote Sens., 13,https://doi.org/10.3390/rs13020197, 2021a.

Dirscherl, M. C., Dietz, A. J., and Kuenzer, C.: Seasonal evolution of Antarctic supraglacial lakes in 2015–2021 and links to environmental controls, The Cryosphere, 15, 5205-5226,https://doi.org/10.5194/tc-15-5205-2021, 2021b.

Li, W., Lhermitte, S., and López-Dekker, P.: The potential of synthetic aperture radar interferometry for assessing meltwater lake dynamics on Antarctic ice shelves, The Cryosphere, 15, 5309-5322,https://doi.org/10.5194/tc-15-5309-2021, 2021.

Wangchuk, S., Bolch, T., and Zawadzki, J.: Towards automated mapping and monitoring of potentially dangerous glacial lakes in Bhutan Himalaya using Sentinel-1 Synthetic Aperture Radar data, Int. J. Remote Sens., 40, 4642-4667,https://doi.org/10.1080/01431161.2019.1569789, 2019.

---

## Author Response (AR1)

Reviewer 1:

***Summary***

*The paper describes a semantic segmentation scheme to map landlocked lakes in Antarctica, using Landsat and Sentinel-1 satellite imagery as base data. Landsat images are segmented with a U-net, Sentinel-1 with a manually tuned threshold. The results are merged with a simple late fusion logic.*

We would like to thank you for reviewing and commenting the manuscript. Below, we present a detailed response to each of your comments, with the original comments in italics and the responses in blue. All the recommended modifications have been implemented in the revised paper.

***Novelty/Relevance***

*There isn't any technical novelty. Methods are standard and used in somewhat ad-hoc manner without clear justification for the design.*

Thanks for your comments. Indeed, we employed the widely used U-Net model to detect landlocked lakes from Landsat images. To overcome the severe cloud interference in the optical images in Antarctic, we further combined the surrounding land cover information of water by using random forest (RF) model from SAR images. Our framework generated the high-resolution products of landlocked lakes' open water (LLOW), which can help us better understand the changes of Antarctic lakes under a changing climate in the future. We revised the manuscript and claimed the novelty of our study as follows:

"What's more, by combining Landsat and Sentinel-1 images, we overcame the severe cloud interference in the optical images in Antarctic, significantly improving the detection frequency of landlocked lakes. We also addressed the challenge of obtaining surrounding land cover information of water in SAR images, thereby successfully generated the high-resolution LLOW products. By providing reliable long-term LLOW series products, our model contributes to a deeper understanding of the dynamic changes of LLOW under a changing climate." (lines 465-470)

In the process of identifying open water with the threshold segmentation method, visually determining the threshold for separating the bimodal histogram appeared ad-hoc. Therefore, we replaced the threshold segmentation with a random forest model, employing machine learning to improve the open water identification. Additionally, we revised some step descriptions to explain the necessity of these steps:

"The RF model, a nonlinear modelling tool, can accurately predict and has a high tolerance to noise and outliers (Huang et al., 2021). We established the RF model for each study area to identify the open water in SAR images according to backscatter and incidence angles." (lines 216-219)

"There are many small LLOW distributed across the four study areas, especially in LHs and VHs, while the U-Net is not ideal for recognizing small-scale open water. Therefore, we resampled the patches with NN from 300*300 pixels to 1024*1024 pixels, in order to magnify the small open water area. After land-cover classification using U-Net, we again resampled these classified results of 1024*1024 pixels to 300*300 pixels with NN. To reduce the border effect caused by U-Net (Dirscherl et al., 2021), we only remained the result of 250*250 pixels in the center of the patch while discarding the edge with a length of 25 pixels." (lines 188-193)

"According to the annotated sample set, some LLOW areas are not within the potential LLOW area range. To leverage the resolution advantage of Sentinel-1 and its potential for LLOW identification, we established a buffer zone for the potential LLOW area (Wangchuk et al., 2019). As shown in Figure S1, the rate of decrease in the ignored LLOW area diminishes as the buffer radius increases. We selected a buffer radius of 20 m, where the reduction in LLOW area is most significant, and resampled the potential LLOW area into a 10-m resolution." (lines 245-249)

*The specific application appears to be new, I am not aware of any paper that described the specific case of land-locked Antarctic lakes. That being said, the distinction is perhaps a tad contrived, there has certainly been work on detecting supra-glacial lakes, so the only difference is really to check whether a lake is surrounded by rock or by ice.*

We agree that the primary difference of detection goals between our work and previous studies is the detection of whether a lake is surrounded by rock or by ice. However, this is a challenging work. Limited by the complexity of rock and ice backscatters, SAR images can't distinguish rock and ice. Due to the interference of clouds, multispectral images are not suitable for generating long time-series datasets. Combining SAR images with multispectral images can solve this problem. We revised the manuscript as follows:

"Unlike the identification of supraglacial lakes, the detection of landlocked lakes requires information of surrounding land covers. Optical remote sensing images are disturbed by frequent clouds in Antarctica, and SAR images have difficulty capturing the information of land covers around lakes. In addition, compared to single-polarization SAR images, the utilization of multi-polarization SAR images can improve the capability to distinguish LLOW from other ground objects (Zakhvatkina et al., 2019). However, the high-resolution GRD products only

provide single polarization over the Antarctic continent. The high-resolution multi-polarization SAR images are not available in Antarctica. Thus, to better understand the dynamics of landlocked lakes in Antarctica, more efficient and accurate methods are needed." (lines 64-70)

***Strengths***

*Since the task has apparently not been studied before, there is potential to systematically map land-locked lakes with the method (it is not done at any scale, though). I am not an expert in Antarctic ecology or climate and cannot judge the relevance of this, but it is a mapping capability that hadn't been investigated.*

*The proposed method works moderately well, even if the segmentation performance is not surprising or spectacular for the fairly straightforward task.*

We would like to thank you for the positive comments on the significance of this work. The threshold method didn't perform very well on the lake detection task, so we have applied the random forest model to identify open water in SAR images to improve the segmentation performance. The LLOW identification model yielded the mean accuracy, F1 score, and mIoU values of 0.94, 0.89, and 0.81 for four study areas on the test set. In addition, the detection of open water in Antarctic is more difficult than in other areas. Interfered by freeze-thaw process, strong wind, and incidence angle (Dirscherl et al., 2021; Li et al., 2021), the backscatters of water is unstable and usually change significantly. We have revised the manuscript as follows:

"The LLOW identification model yielded the mean accuracy, F1 score, and mIoU values of 0.94, 0.89, and 0.81, respectively, for four study areas on the test set. We further validated the model performance on four test patches (Fig. 8). The LLOW identification model yielded the F1 scores ranging from 0.88 to 0.95 and mIoU ranging from 0.81 to 0.90." (lines 331-334)

***Weaknesses***

*Technical decisions seem somewhat arbitrary and ad-hoc. Not seriously wrong, but the described scheme is just "a way to do it", not a carefully designed and justified "best way to do it".*

Thanks for your suggestion on our workflow. We have revised the representation of technical steps to clarify why we conducted them. For example, we explained the necessity of resampling patches from 300*300 to 1024*1024 and masking SAR images with Landsat images in the pre-processing section and post-processing section respectively.

"There are many small LLOW distributed across the four study areas, especially in LHs and VHs, while the U-Net is not ideal for recognizing small-scale open water. Therefore, we resampled the patches with NN from 300*300 pixels to 1024*1024 pixels, in order to magnify the small open water area." (lines 188-190)

"The use of Landsat images in the visible and near-infrared bands is significantly hindered by cloud interference, especially along the Antarctic coast. As mentioned in Section 2.2, within the four study areas over 2014-2022, only a total of 79 Landsat images are suitable for LLOW detection. Therefore, the number of Landsat images with low cloud cover in the study areas is insufficient for our time series analysis." (lines 231-234)

*The evaluation is rather weak, using only a few small areas, and even excluding some lakes that are clearly visible within the image tiles. The study does not go beyond the four small proof-of-concept regions, there are no large-scale, wall-to-wall results.*

Based on your suggestions, we have increased the size of the sample set as much as possible. We conducted visual interpretation for Sentinel-1 images with the assistance of Landsat images (Liang and Liu, 2020). Since the dates of the Landsat images and Sentinel-1 images must be close to each other and cloud-free Landsat images are scarce, we used all suitable Landsat images and annotated 46 patches with the 300*300 size to train the random forest model and validate the model accuracy. We evaluated the accuracy of LLOW identification using these 46 patches, calculating the accuracy, F1 score, and mIoU of the results. Additionally, we reported the confusion matrices and spatial error distribution for four patches. In Antarctic, the regions with landlocked lakes are rare and small, and the landlocked lakes themselves are also of small size. Mapping the entire SAR images to display the wall-to-wall results would obscure these landlocked lakes, especially in LHs. Therefore, our final results focus on the areas where landlocked lakes are present within the full SAR images. We revised the manuscript as follows:

"The LLOW identification model yielded the mean accuracy, F1 score, and mIoU values of 0.94, 0.89, and 0.81, respectively, for four study areas on the test set. We further validated the model performance on four test patches (Fig. 1). The LLOW identification model yielded the F1 scores ranging from 0.88 to 0.95 and mIoU ranging from 0.81 to 0.90. Among the four areas, SO exhibited the highest mIoU value of 0.90, suggesting the most similar spatial distribution between the predicted LLOW and the ground truth. LHs showed the lowest mIoU of 0.81, while CWM and VHs showed the mIoU values of 0.82 and 0.83, respectively. In VHs and SO, the locations and areas of LLOW were well recognized (Figs. 1k and 1l). In LHs, the spatial distribution of LLOW was also accurately detected, although there were some inconsistencies in the boundaries between the ground truth and the predicted lakes (Fig. 1j). In addition, in CWM, the model successfully identified all LLOW areas, but it misclassified the areas covered by floating ice with low

backscatter (Fig. 1a) as LLOW. Overall, our model demonstrated proficiency in detecting LLOW areas, providing reliable information on the spatial distribution and extent of LLOW.

[Figure]

Fig 1. Validation of landlocked lake identification model in testing dataset for four areas. The four columns of images are validation images for CWM, LHs, VHs and SO. The first, second, and third rows are ground truth, predicted and spatial errors images, respectively. The background images are displayed from false color combination of 7-4-3 bands. The spatial distribution of classification errors is obtained from overlapping ground truth and predicted images." (lines 331-346)

*The model validation suggests that almost all the performance is due to Landsat, whereas Sentinel-1 does not offer much except the potential to densify in time - which however is not actually done, since the Landsat segmentation acts as a hard constraint: the algorithm does not appear to allow SAR to add lake pixels.*

We agree that the potential LLOW areas strictly constrain the potential of SAR images in the current algorithm. We established a buffer around the potential LLOW areas with a 20-m buffer radius (Wangchuk et al., 2019) to allow water to extend beyond previous masks. We revised the manuscript as follows:

"According to the annotated sample set, some LLOW areas are not within the potential LLOW area range. To leverage the resolution advantage of Sentinel-1 and its potential for LLOW identification, we established a buffer zone for the potential LLOW area (Wangchuk et al., 2019). As shown in Figure S1, the rate of decrease in the ignored LLOW area diminishes as the buffer radius increases. We selected a buffer radius of 20 m, where the reduction in LLOW area is most significant, and resampled the potential LLOW area into a 10-m resolution. After that, we combined the Landsat and Sentinel images, using the potential extents of LLOW and the open water derived from SAR, to generate the long-term series of LLOW." (lines 245-250)

*Finding of a "decreasing trend in LLOW area" for 2017-2021 is rather trivial and expected. It would be more interesting to interpret the measured areas beyond just that obvious trend.*

We thank you for the insightful comment. The reliance on available satellite imagery inherently limits the scope of our trend analysis. Despite our efforts to utilize the most comprehensive and up-to-date datasets, the frequency and resolution of the images available for analysis limited our ability to identify and interpret more complex trends within the LLOW dynamics over the study period. Specifically, the temporal resolution of the imagery made it challenging to capture short-term fluctuations or subtle changes in LLOW area that could potentially reveal more about the underlying processes affecting these water bodies.

**Presentation**

*Throughout, the text could be made shorter and more concise. E.g.,*

- *lines 215-220 are unnecessary, everything that is said there is already implied by the use of U-net*

Thanks for your suggestions. We have removed these sentences.

- *228-240 is a verbose, meandering way to simply say "we manually chose a global threshold by inspecting histograms"*

Thanks for your suggestions. We replaced the thresholding method with a random forest to identify open water in SAR images and removed the paragraph about the thresholding method.

- *250-264 says little more than that the definition of a land-locked lake is a water region surrounded by a rock region.*

Thanks for your suggestions. In this paragraph, we describe the process of BFS in detail and explain how a land-locked lake is identified. We have shortened this paragraph as follows:

"A landlocked lake is a water region surrounded by a rock region. Not all "open water" pixels extracted through the open water identification models are LLOW, such as glacial rivers and melted water from coastal glaciers. Besides, LLOW may be indirectly surrounded by rocks. For example, LLOW may be enclosed by ice, which in turn is surrounded by rocks. In our classified results, a classified Landsat image consists of a connected non-rock area and interspersed rock areas containing LLOW. The BFS algorithm has been proven to be effective in removing the connected areas (Silvela and Portillo, 2001). Thus, the BFS algorithm can effectively eliminate the connected non-rock area while retaining the rock areas. BFS simulates the spreading of seawater in the Antarctic summer and leaves only rock areas where stable LLOW may exist. The supraglacial lakes, epiglacial lakes, and seawater are all removed during BFS. Finally, all the remaining open water pixels derived from Landsat images are extracted and marked as "LLOW".
" (lines 222-230)

- *etc.*

We further shortened the method section. For example, we moved the process of sample set preparation to Section 2.2 (Dataset). Consequently, we removed the content about sample annotation from Section 3.1 and Section 3.4, in order to focus more on data pre-processing and model validation.

*The introduction is verbose and not very focussed, touching on all sorts of studies about land-locked lakes that have no relation or importance for what the paper then does.*

Thank you very much for your comment on the introduction part. We have shortened the paragraph about microorganisms in Antarctic lakes in the introduction as follows:
"Extensive research confirms diverse microorganisms in Antarctic lakes, including prokaryotes like bacteria and eukaryotes such as phytoplankton (Parnikoza and Kozeretska, 2019; Izaguirre et al., 2021; Keskitalo et al., 2013; Rochera and Camacho, 2019). Cyanobacteria play a crucial role in primary production and nutrient cycling, as highlighted by studies on their diversity and distribution (Taton et al., 2006; Komárek et al., 2012), alongside findings on unique microbial assemblages, such as Hymenobacter sp., and diverse bacterial communities (Koo et al., 2014; Huang et al., 2014; Carvalho et al., 2008; Papale et al., 2017). These studies underscore the ecological importance and high diversity of Antarctic lake

ecosystems." (lines 31-36)

*The analysis in lines 420-440 is rather hand-wavy, I was not able to see what purpose it actually serves. It gives me the impression that the authors just performed a random analysis that was easily doable, to send a message that the maps could potentially serve some useful purpose.*

We appreciate your feedback regarding the analysis presented in lines 420-440. The purpose of this section was to elucidate the complex relationship between Positive Degree Days (PDD) and the area change of landlocked lake open water (LLOW) in different areas. We recognize the reviewer's concern that the analysis may appear cursory or lacking in clear purpose. To address this, we offer the following clarifications and enhancements to underscore the relevance and rigor of our analysis.

The primary purpose of the analysis was to demonstrate the significant influence of PDD on LLOW area changes over time. By showing a strong correlation (average R2 value of ~ 0.9) during the growth phase of LLOW areas, we aimed to validate PDD as a key factor explaining most of the variability in LLOW area.

We further found that PDDs impact year-to-year LLOW area fluctuations non-linearly. Our revised paragraph is as follows:
"PDDs can also influence the year-to-year fluctuations in LLOW area, but the relationship between changes in PDD and LLOW area is non-linear. For instance, the maximum PDD in 2017 was more than 2 times higher than that in 2018 in LHs, yet the maximum area increased by 50% (Figs. 10a and 10b). The maximum area of VHs remains relatively stable over the 5 years. When PDD reaches a certain threshold, all LLOW areas have already melted, so further increases in PDD do not lead to changes in LLOW area. Therefore, across different years, significant differences in PDD can result in minimal variation in LLOW areas. Based on this, it can be inferred that the threshold for PDD in LHs is likely between 25°C and 35°C. When PDD exceeds 35°C, the maximum LLOW area keeps relatively invariant at ~ 0.5 km$^2$." (lines 389-395)

*There are remaining language issues, both in terms of English grammar (random example: "due to non-uniform of field surveys") and in terms of technical expressions (e.g., "gradient disappearance" instead of "gradient vanishing").*

We appreciate the reviewer's comments. We have revised the language and technical expressions within our manuscript.

***Technical Questions***

*The computational procedure is not entirely clear. It is one way of cobbling together a segmentation pipeline, but there is no clear explanation why that specific design was chosen. Whereas there are obvious concerns about it, e.g.,*

- *the potential benefit of Sentinel-1 for the land-cover map are not exploited*

We have established the buffer regions for the potential LLOW area, allowing Sentinel-1 images to add LLOW pixels in the identification model. We revised the manuscript as follows:

"According to the annotated sample set, some LLOW areas are not within the potential LLOW area range. To leverage the resolution advantage of Sentinel-1 and its potential for LLOW identification, we established a buffer zone for the potential LLOW area (Wangchuk et al., 2019). As shown in Figure S1, the rate of decrease in the ignored LLOW area diminishes as the buffer radius increases. We selected a buffer radius of 20m, where the reduction in LLOW area is most significant, and resampled the potential LLOW area into a 10-m resolution." (lines 245-250)

- *possible correlations between optical and SAR are lost*

The optical and SAR images with dates close to each other exhibited high correlations for land-cover classification. However, due to the lack of cloud-free Landsat images in Antarctica, combining optical and SAR images with similar dates reduces the number of suitable images, leading to a lower temporal resolution. Thus, we aggregate all Landsat images into one potential LLOW area instead of the fusion of each Landsat and Sentinel-1 image.

- *the fusion seems to not leverage away the (pseudo-)probabilistic segmentation scores*

To combine Landsat and Sentinel-1 images using the probabilistic segmentation scores, we attempted to incorporate the potential LLOW areas as an input feature in the random forest model for directly identifying LLOW instead of masking. However, the trained model misclassified the snow-covered areas with high backscatter within the potential LLOW areas and seawater areas with low backscatter as LLOW. Thus, we finally decided to apply the random forest model for open water identification and mask the SAR results.

*What is meant by saying 300x300 is the "common" patch size for U-net models? There isn't a single, canonical patch size for training those models, and at test time they are anyways fully convolutional and not tied to a specific patch size.*

Thanks for your suggestion. This is an inaccurate description. We have deleted this word.

*I don't understand the upsampling of the input for the U-net. No information is added by this and the effective receptive field / context window inside the network is actually reduced. So it would seem that one can reach at least equal performance, with lower computational effort, by properly training the U-net to handle the smaller images. Please explain.*

The 1024*1024 patches consume more computational resources than 300*300 patches during the U-Net inference. However, the upsampling is used to magnify the small open water areas. Some open water areas only consist of a few pixels in the images. Magnifying these areas through upsampling contributed to the identification of small LLOW. We clarified the description as follows:

"There are many small LLOW distributed throughout four study areas, especially in LHs and VHs, while the U-Net is not ideal for recognizing small-scale open water. Therefore, we resampled the patches with NN from 300*300 pixels to 1024*1024 pixels, in order to magnify the small open water area. After land-cover classification using U-Net, we again resampled these classified results of 1024*1024 pixels to 300*300 pixels with NN." (lines 188-191)

*Certain augmentations should be ablated and empirically justified. Conceptually it seems problematic to apply transformations like rotating or vertical flipping, as this leads to illumination directions that are implausible in real Landsat images, especially at Antarctic altitudes with low sun elevation.*

Thanks for your suggestion. We augmented the datasets with only translation operations and trained the U-Net. The new U-Net performs similarly to the previous model. The land-cover identification with U-Net may not depend on the direction of sunlight incidence. Therefore, we continued to use the previous augmentation method.

*Using a threshold for segmentation is of course entirely correct and sensible, if it works. But the justification that single-channel input will lead to "instability" of U-net makes no sense. Countless applications use U-nets with various single-channel inputs (SAR, panchromatic, depth,...).*

Thanks for your suggestion. This is an inaccurate description. We have deleted the sentence: "The single-channel input from Sentinel-1 images will lead to significant instability in the results of the U-Net model."

*The fusion step is unclear. First you argue for using SAR, and for combining it with optical data, to obtain better temporal resolution. But then, a consensus over at least 2 Landsat acquisitions is required for a potential LLOW pixel, meaning that the shortest possible resolution of everything that follows is the interval between three Landsat overpasses (if a pixel flips from ice to water between two consecutive images,*

*you need to wait for a third image to confirm, so over the entire period you cannot say whether the pixel remained the same or thawed and froze again).*

We used all land-cover maps during five years to generate one potential LLOW area map for one region. Among several dozen images, if a pixel is identified as LLOW in at least two images, then that pixel is considered a potential LLOW. This algorithm aims to preserve as much potential space as possible. We revised the manuscript as follows:

"Specifically, if a pixel was identified as LLOW two or more times from 2014 to 2022, it was considered as a potential LLOW pixel. We aggregated all LLOW distribution images and obtained one potential LLOW area for each study area." (lines 243-244)

*In line 290, it remains unclear how the authors "disregard" underestimated lakes. To do that one must identify them first - but the algorithm, by definition, does not know where it made an underestimation error.*

We manually deleted those images which underestimated the LLOW. In the melting phase, the backscatters of water may suddenly increase due to strong wind or other reasons (Fig. 2). We will check the images where the lake areas suddenly decrease in the time series curve. If backscatters of water increase abnormally, we will manually remove these images.

[Figure]

Figure 2. The significant backscattering changes of lakes during 25 January, 31 January, and 6 February in the VHs.

*I do not understand why only a tiny set of 17k pixels were "annotated for U-net", but 225k pixels were "annotated for LLOW identification". 17k seems an overly small training set: assuming an average lake size of, say, 600x600m that would be fewer than 50 lakes. Why would one do that if apparently another 225k annotated pixels are available?*

The labels annotated for U-Net and LLOW identification are different. The labels for U-Net consist of water, rock and ice, while the labels for LLOW only consist of LLOW and non-LLOW. Thus, we can't apply LLOW data to U-Net testing.

*Cohen's kappa as a segmentation metric is discouraged (cf. [Pontius and Milliones, 2011]). I would recommend to follow best practice and show confusion matrices, F1 scores, IoU scores.*

Thanks for your suggestion. We have removed the content about kappa and added new accuracy reports about accuracy, F1 scores, and mIoU as follows:

"The accuracy of classification models is estimated by confusion matrix, accuracy, F1 score and mean IoU. The formulas are presented in Eqs. (1), (2), (3), (4), and (5).

$$\text{Accuracy} = \frac{TP}{TS} \tag{1}$$

$$F1 = 2 * \frac{\text{Precision} * \text{Recall}}{\text{Precision} + \text{Recal}} \tag{2}$$

$$\text{Precision} = \frac{TP}{TP + FP} \tag{3}$$

$$\text{Recall} = \frac{TP}{TP + FN} \tag{4}$$

$$\text{mIoU} = \frac{1}{N} \sum_{i=1}^{N} \frac{TP_i}{TP_i + FP_i + FN_i} \tag{5}$$

where N is the number of categories; TS is the total number of samples; TP is the number of true positive classified results; FP is the number of false positive classified results; TN is the number of true negative classified results; and FN is the number of false negative classified results in confusion matrix." (lines 265-273)
"The LLOW identification model yielded the mean accuracy, F1 score, and mIoU values of 0.94, 0.89, and 0.81, respectively, for four study areas on the test set. We further validated the model performance on four test patches (Fig. 8). The LLOW identification model yielded the F1 scores ranging from 0.88 to 0.95 and mIoU ranging from 0.81 to 0.90." (lines 331-334)

*The PDD metric (equation 4) is defined in a strange manner. According to the definition that metric is exactly the same for a cold spell with two weeks of constantly at 0 degrees, or of constantly -25 degrees. Surely that would make a difference for the ice cover of the lakes? Wouldn't it be more natural to look at the number of consecutive PDD, or to integrate the average temperature including negative values?*

We thank you for the insightful comments and suggestions. The use of the PDD metric in our study is based on its common application in glaciological and climatological research as a simplified measure to estimate the melting potential of ice and snow. The rationale behind focusing exclusively on temperatures above 0°C is to directly quantify the thermal energy available for melting ice, which is a critical factor in the dynamics of lake ice cover.

However, you raised an important point about the continuity of temperature conditions during the melting phase. It is essential to highlight that temperature patterns preceding the accumulation of positive degree-days (PDDs) typically show a gradual increase towards 0°C. This pattern means that, even before reaching the threshold for PDD calculation, the thermal conditions are already near the melting point. Thus, it is hardly possible to have a cold spell with two weeks of constantly at -25 degrees during the melting phase.

To assess the influence of higher temperatures more accurately on the expansion of open water areas, we utilized the average daily temperature, inclusive of negative values, to calculate PDDs. This method acknowledges that any day with an average temperature above 0°C contributes to the melting process, providing a nuanced measure of thermal energy input relevant to ice melt. However, days with an average temperature below 0°C are considered to have negligible impact on ice melting and are thus not included in our PDD calculations.

*For the decline (Section 5.2), why use the minimum temperature? To my knowledge, and also more in line with the PDD metric used earlier, a more common indicator is the number of consecutive negative degree days, at least in studies of lake ice in Canada, the Alps, etc.*

We appreciate your comment. We acknowledge the commonality of using consecutive negative degree days (NDDs) in lake ice studies across regions such as Canada and the Alps, which serves as a direct measure of the cold spell's duration and intensity influencing ice formation and maintenance.

In the revised manuscript, we employed NDDs instead of temperature to analyze the decline phase of lake ice cover. Across all four study regions, we found R-squared values exceeding 0.5, demonstrating that NDDs effectively explain the

reduction in LLOW area during the freezing period. Additionally, in certain specific cases, temperature may better explain fluctuations in the LLOW area, such as instances of temperature-driven rebounds during periods of decline. We revised the manuscript as follows:

"Cumulation in successive negative air temperature days contributes to the lowering of water temperature and the commencement of the water freezing process, i.e., the formation and longer-term persistence of ice cover (Graf and Tomczyk, 2018). Therefore, we calculate the negative degree-day sum (NDD) by using Eq. (7), which represents the cumulative sum of temperatures below the melting point during a specific period.

$$NDD_n = \sum_{i=0}^{n} \begin{cases} T_i, & T_i < 0 \\ 0, & T_i \geq 0 \end{cases} \tag{7}$$

Here, the negative degree-day sum prior to the day n is denoted as $NDD_n$ (°C) and $T_i$ represents the station mean temperature (°C) measured on day i. The relationship between the LLOW area and NDD in each area during the freezing season is significant (Table 1). The calculation of the $R^2$ value was based on a linear fit of the NDD and the LLOW area, ranging from the maximum LLOW area to the minimum. In all four study areas, the $R^2$ values were found to be greater than 0.5. This indicates a strong response of the LLOW area to NDD changes during the decline phase of the LLOW area.

Table 1. $R^2$ of the LLOW area and NDD in freezing seasons between 2017 and 2021. The average is the derived from the mean values of $R^2$ for each area across these years.

| Year | CWM | LHs | VHs | SO |
|---|---|---|---|---|
| 2017 | 0.52** | 0.95** | 0.72** | |
| 2018 | 0.84** | 0.94** | 0.85** | |
| 2019 | 0.73** | 0.75** | 0.82** | |
| 2020 | 0.88** | 0.85** | 0.78** | 0.78** |
| 2021 | 0.57** | 0.59** | 0.86** | 0.97** |
| Average | 0.71 | 0.82 | 0.81 | 0.88 |

* $p < 0.05$, ** $p < 0.01$." (lines 405-416)

***Minor Comments***

*Why do ice-covered lakes "magnify the warming trend"? I would think they might rather dampen it?*

We appreciate the comment. Ice and snow typically have high albedo, reflecting a significant portion of incoming solar radiation, thus potentially having a cooling

effect on the local environment. However, the context of our discussion focuses on the transition period from ice-covered to ice-free conditions and the consequent alterations in surface albedo and energy absorption.

The mechanism by which ice-covered lakes can amplify warming trends is linked to the concept of the ice-albedo feedback. During the winter and early spring, ice-covered lakes indeed reflect a large portion of solar radiation due to their high albedo. However, as temperatures rise, ice begins to melt, reducing the surface area covered by ice and exposing the underlying water. Liquid water has a significantly lower albedo than ice, indicating that the lake absorbs more solar radiation instead of reflecting it. This increased absorption of solar energy by the water surface not only contributes to further warming and melting of the remaining ice but also leads to a rise in water temperature, which can enhance local warming. This mechanism aligns with observations and modeling studies that have documented the significant role of ice-albedo feedback in accelerating warming, particularly in polar and high-latitude regions where ice and snow cover are integral to the climate system.

*It is a strange claim that U-net "requires less training datasets and time" than other neural networks. That depends on who you compare to, of course there are designs that are faster than U-net (e.g., those created for mobile or embedded devices). Moreover, "U-net" isn't a specific architecture but a whole family of networks with certain characteristics - essentially, symmetric hourglass encoder-decoder structure with dense skip connections. So some "U-nets" are a lot slower and more data-hungry than others.*

Thanks for your suggestion. We have revised the manuscript as follows:

"U-Net neural network is a deep learning network for semantic segmentation based on a fully convolutional network (Ronneberger et al., 2015), which is faster to train due to its context-based learning approach (Siddique et al., 2021)." (lines 196-197)

Reviewer 2:

**Summary**

*The manuscript aims to detect landlocked lakes in Antarctica fusing optical and SAR imagery and using a U-net based method.*

We would like to thank the Referee 2 for reviewing and commenting the manuscript. Below, we present a detailed response to each of your comments, with the original comments in italics and the responses in blue. All the recommended modifications have been implemented in the revised paper that will be uploaded.

**Novelty/Relevance**

*I'm not aware of another method addressing land-locked Antarctic lakes. However, the methods used are standard methods, or in the case of thresholding the SAR imagery outdated within the field of research. The thresholding method also means that the lakes under different wind states can't be separated. Moreover, the method can't separate these lakes from other types of lakes, not surprising, but if that was the goal the method needs to be further improved.*

Thanks for your suggestion. We have replaced the threshold segmentation with a random forest model, employing machine learning to improve the open water identification. In the random forest model, we considered both backscatter and incidence angles to detect open water. We revised the manuscript as follows:

"The RF model, a nonlinear modelling tool, can accurately predict and has a high tolerance to noise and outliers (Huang et al., 2021). We established the RF model for each study region to identify the open water in SAR images according to backscatter and incidence angles." (lines 216-219)

**Strengths**

*Study of land-locked lakes in Antarctica, where the method is designed to detect a specific type of lakes. Could this method be applied to other landlocked lakes to monitor water resources? If so then it might become a more viable method.*

Thanks for your suggestions for the potential application. We trained the U-Net model to adapt different conditions using various training data, such as thin clouds, mountain shadows and floating ice, which leads to the robust workflow for other study areas. Therefore, our method has the potential to perform well in other regions, such as identifying the other landlocked lakes in Antarctica or detecting numerous landlocked lakes along the coastal areas of Greenland. What's more, the BFS algorithm is able to distinguish open rivers and closed lakes. It

suggests that our method could identify landlocked lakes from sea water and rivers within an image on the Alaskan North Slope. Consequently, we can detect the growth of thermokarst lakes and their integration with river drainages. We clarified the potential application of our methods as follows:

"Although the LLOW identification model has these limitations, our findings demonstrate its strong performance across the four study areas. The deep learning approach, namely U-Net, enhanced model robustness across diverse environmental conditions such as various surrounding features, cloud covers, lighting conditions, and mountain shadows. Using the RF model to identify open water in SAR images can also overcome unstable factors such as cloud cover, producing the stable high-resolution time series of open water. Therefore, our method has the potential to perform well in other regions, such as identifying the other landlocked lakes in Antarctica or detecting numerous landlocked lakes along the coastal areas of Greenland. Additionally, our proposed method for distinguishing between seawater, supraglacial lakes, and landlocked lakes can be applied to the identification of thermokarst lakes, such as the numerous thermokarst lakes on the Alaskan North Slope. The BFS algorithm can distinguish between open rivers and closed lakes on plain permafrost. By utilizing BFS and the fusion of Landsat and Sentinel-1 images, we can differentiate thermokarst lakes and river drainages within an image. Consequently, the growth of thermokarst lakes and their integration into river systems can also be detected." (lines 455-465)

*Weaknesses*

*The manuscript is weak in the technical details, in particular there is an apparent lack of understanding of satellite images and details around them are missing. How is the SAR data pre-processed? Is the different spatial resolution between the optical and SAR images accounted and corrected for? The incidence angle dependency in SAR data will result in higher incidence angles having a lower backscatter response. How is this accounted for in the method? Are only repeat orbits used? How will the incidence angle affect the results here?*

Thanks for your suggestions for the pre-process of SAR images.

1) We have modified the pre-processing operations of the Sentinel-1 level-1 GRD data. We conducted the orbital correction, thermal noise removal, radiometric calibration, speckle filtering, terrain correction and decibel conversion on the Sentinel-1 Level-1 GRD products using ESA's Sentinel Applications Platform (SNAP) software. We revised the manuscript as follows:

"For Sentinel-1 images, we performed orbital correction, thermal noise removal, radiometric calibration, speckle filtering, terrain correction and decibel conversion on the Sentinel-1 Level-1 GRD products using ESA's Sentinel Applications Platform (SNAP) software. Then the corrected Sentinel-1 images were then reprojected and

cropped to align with the spatial extent of the cropped Landsat images." (lines 174-178)

2) The spatial resolution of Sentinel-1 images is 10 m. Landsat images maintained at a 30-m resolution until they are resampled to 10-m resolution after being processed into potential LLOW areas. We revised the manuscript to describe the resampling process:

"We selected a buffer radius of 20 m, where the reduction in LLOW area is most significant, and resampled the potential LLOW area into a 10-m resolution." (lines 248-249)

3) Considering the pre-process method proposed by Wangchuk et al. (2019), we used the same descending orbits to reduce the influence of incidence angles. What's more, we established the random forest model to identify the open water in SAR images according to backscatter and incidence angles. We revised the manuscript as follows:

"All Sentinel-1 images are from the descending orbit, in order to avoid geometric distortions and orthorectification limitations (Wangchuk et al., 2019)." (lines 117-118)

"We established the RF model for each study region to identify the open water in SAR images according to backscatter and incidence angles." (lines 217-219)

4) The incidence angles play an import role in the random forest and yield a high feature importance. We discussed it as follows:

"However, there is no obvious linear correlation between LLOW backscatter and wind, incidence angles or slope. What's more, we added the wind speed, incidence angles, and slope as input features for the RF model in open water identification. However, only incidence angle yields a significant feature importance. This indicates that the incidence angle is much more important for open water detection compared to wind speed and slope. Thus, our RF model did not consider the wind features and slope." (lines 445-449)

*Radar shadows (e.g. mentioned on P12 R271) are a well-known issue within SAR images. A method should be designed to deal with them or at least quantify the scale of the issue.*

Thanks for your suggestion. Because the steep terrain yields mountain shadows and identification errors (Dirscherl et al., 2021), we calculated the slopes from DEM to evaluate the influence of radar shadows. As shown in Figure 1, there is no significant linear correlation between LLOW backscatter and slope. Thus, the

impact of radar shadows on the identification results is minimal. We discussed this issue as follows:

"Because the steep terrain yields mountain shadows and identification errors (Dirscherl et al., 2021), we calculated the slopes from DEM to evaluate the influence of topography. To evaluate the influence of wind, incidence angles, and topography, we sampled within the LLOW areas of four study areas from 2017 to 2021 from the 46 sample patches (Figs. S3, S4, S5 and S6). However, there is no obvious linear correlation between LLOW backscatter and wind, incidence angles or slope." (lines 442-446)

[Figure]

Figure 1. The relationship between mean backscatter of LLOW and slopes. LLOW and slope samples came from the 46 annotated sample patches. There is no significant correlation between mean backscatter and slopes.

*Wind may cause a wind roughened (high backscatter) surface, it appears that the model can only deal with low backscatter surface scattering surfaces. The method would then only be applicable in a limited number of SAR images and this limitation would hinder an operationalization or a processing chain with a larger number of images. Moreover, separation of open water areas from surrounding is challenging due to the varying backscatter values under different wind conditions. To make a method applicable to be used in an ML/DL/operational setting all wind states need to be accounted for in the method. Something that is challenging for a threshold-based method.*

Thanks for your suggestion on the impact of wind. We evaluated the influence of wind on the LLOW backscatter (Figures 2 and 3 below). However, there is no

significant linear correlation between wind and LLOW backscatter. In addition, we tried to add wind speed as the input feature for the random forest in open water identification. The contribution fraction of wind speed is less than 1%, so we didn't consider the wind features in our random forest model. We revised the manuscript as follows:

"The backscatter of LLOW is mainly disturbed by two types of factors: the first is external factors, such as wind speed and direction, SAR image incidence angles, and mountain shadows; the other is the LLOW surface cover, such as floating ice and snow. Firstly, when the open water is disturbed by wind, the backscatter increases. Additionally, the incidence angles and topography also affect the backscatter of open water. Because the steep terrain yields mountain shadows and identification errors (Dirscherl et al., 2021), we calculated the slopes from DEM to evaluate the influence of topography. To evaluate the influence of wind, incidence angles, and topography, we sampled within the LLOW areas of four study areas from 2017 to 2021 from the 46 sample patches (Figs. S3, S4, S5, and S6). However, there is no obvious linear correlation between LLOW backscatter and wind, incidence angles or slope. What's more, we added the wind speed, incidence angles, and slope as input features for the RF model in open water identification. However, only incidence angle yields a significant feature importance. This indicates that the incidence angle is much more important for open water detection compared to wind speed and slope. Thus, our RF model did not consider the wind features and slope." (lines 439-449)

[Figure]

Figure 2. The relationship between mean backscatter of LLOW and wind features. LLOW samples came from the 46 annotated sample patches. The wind_u and wind_v are derived from ERA5 dataset, representing the wind velocity in the east

and north direction, respectively. There is no significant correlation between mean backscatter and wind speed or direction.

[Figure]

Figure 3. The relationship between mean backscatter of LLOW and wind speed. LLOW samples came from the 46 annotated sample patches. The wind speed is calculated from wind_u and wind_v. There is no significant correlation between mean backscatter and wind speed.

*The text should be significantly shortened to avoid unnecessary repetitions, focus the message on what was done here (without repetitions). E.g. among other things can section 3.4 be significantly shortened by removing repetitions. As the other reviewer has already pointed out that the text is verbose and provided examples I'll not do so further here.*

Thanks for your suggestion. We have removed the repetitive sentences and shortened the section 3.3 and 3.4.

*Stating that "best" analysis etc has been used should be strengthened to indicate what makes this the "best".*

Thanks for your suggestion. The U-Net model shows good performance across various terrains and conditions. For example, it is able to overcome the interference from the clouds, shadows, and floating ice. The robust U-Net model contributes to the generation of potential LLOW areas for the entire Antarctica. The thresholding method also performs similarly across all regions. Thus, we stated that our workflow for LLOW identification is robust and can be applied to different Antarctic areas. We revised this point as follows:

"The deep learning approach, namely U-Net, enhanced model robustness across diverse environmental conditions such as various surrounding features, cloud covers, lighting conditions, and mountain shadows. Using the RF model to identify open water in SAR images can also overcome unstable factors such as cloud cover, producing the stable high-resolution time series of open water area. Therefore, our method has the potential to perform well in other regions, such as identifying the other landlocked lakes in Antarctica or detecting numerous landlocked lakes along the coastal areas of Greenland." (lines 456-461)

*The correlation between PDD and lake area is long established and is not new, and neither is different lake shape evolution with different PDD evolution. The fact that lakes melt first from the edges is fundamental knowledge and not new knowledge established here. Combined can these results easily be referred to in existing literature, and this manuscript should highlight what is new knowledge aside from these well-established results.*

We appreciate your insights regarding the established knowledge. We acknowledge that these concepts are well-established within the scientific community and are not presented as novel findings within our manuscript. In the original version of the manuscript, we simply provided a comprehensive background. We will remove these points in the revised manuscript.

*Figure 9 show significant amount of lake area before the start of the study, in order to show time series of lake evolution at least for 3 of the sites the time series needs to be expanded to include data from at least one month earlier. How is the time series affected by removing all the troublesome images, e.g. the wind affected and those where the method failed? Lack of useful data at the start and end of the season will lead to under/over estimation of the lake area. Can the method (time series) be said to satisfactory deal with rapid changes? Or is there a need to increase sampling frequency? P20R402. How does the lack of data in December affect this? It appears that for at least the CWM side data from earlier months is needed to establish maximum lake area and probably also the LH site judging from Figure 9.*

1) Due to floating ice or other factors, large areas of backscatter increase often occur in lakes. The backscatter of the entire lake can rise to a level similar to the surrounding rocks or ice, making them indistinguishable. This phenomenon can lead to a decrease in the identified LLOW area by 10% to 50%. Therefore, when we remove these images, the time series become much more stable.

2) We agree that lack of valid data at the start and end of the season will lead to less understanding about minimum LLOW areas in frozen state. However, due to the 6-day interval between consecutive Sentinel-1 images, the LLOW time series can capture the maximum of LLOW areas. Thus, the current sampling frequency is sufficient.

3) The persistently high LLOW areas in the CWM from January to February indicate that the area in January has already reached the maximum extent of the lakes. The fact that the lakes won't freeze in January suggests that the area in December is expected to be smaller than that in January. Therefore, the absence of data for December does not lead to an underestimation of the maximum lake area.

*There are 3 different lakes presented here, how can this method separate the three types if they all exist in one satellite image?*

These three types of lakes, supraglacial lakes, epiglacial lakes, and landlocked lakes, have a distinguishing characteristic. Supraglacial lakes are surrounded by ice rather than the rock. Epiglacial lakes have only a partial contact with rocks but is not within a rocky area. Landlocked lakes are entirely situated within a rocky area. After land cover classification with the U-Net model, BFS can distinguish their relative positions to rocks in one image. The positional information of LLOW obtained from Landsat can be used as a reference for identifying the location of LLOW in Sentinel-1 images. Finally, landlocked lakes can be extracted from other types of lakes in Sentinel-1 images. We revised the manuscript to clarify this:

"BFS simulates the spreading of seawater in the Antarctic summer and leaves only rock areas where stable LLOW may exist. The supraglacial lakes, epiglacial lakes, and seawater are all removed during BFS. Finally, all the remaining open water pixels derived from Landsat images are extracted and marked as "LLOW"." (lines 227-230)

*P13R290-292. If the LLOW are underestimated who does this affect the biological component that has been used as an argument for conducing the entire study?*

We appreciate your attention to this. If LLOW are underestimated, it could lead to a conservative assessment of available habitats for various aquatic species, which in turn could affect our understanding of biodiversity, ecological interactions, and the potential for conservation efforts within these ecosystems. Such underestimation might also impact the accuracy of ecological modeling and predictions regarding the distribution and abundance of species, which are crucial for formulating effective conservation strategies and understanding ecosystem dynamics in coastal Antarctica.

*P14R317. It is stated that using the thresholding method produces large amount of errors. Establishing an improved method should therefore be a goal of this manuscript. Open water areas that are either sea water, melt lakes on the ice sheet or lakes on land is not possible using simple thresholding in the SAR images as the radar signal interprets each as water.*

Thanks for your suggestion. The interpretation of SAR imagery is challenging due to ambiguous backscatter returns and image geometry effects (Li et al., 2021). Due

to technological limitations and the scale of training datasets, we were unable to implement the deep learning for water identification with SAR images. However, we replaced the thresholding method with the random forest and significantly improved the identification accuracy as follows:

"BFS simulates the spreading of seawater in the Antarctic summer and leaves only rock areas where stable LLOW may exist. The supraglacial lakes, epiglacial lakes, and seawater are all removed during BFS. Finally, all the remaining open water pixels derived from Landsat images are extracted and marked as "LLOW"." (lines 227-230)

*P21R411-420. Lake growth after temperatures start exceeding 0 has been well studied on the Greenland ice sheet for well over 10 years now. And PDD was used by, e.g. Johansson et al, (2013) to study lake evolution.*

*Johansson, Jansson and Brown (2013): Spatial and temporal variations in lakes on the Greenland Ice Sheet, J. Hydrology, https://doi.org/10.1016/j.jhydrol.2012.10.045*

We greatly appreciate the reviewer's insightful comment and the reference to the seminal work of Johansson et al. (2013) on the spatial and temporal variations in lakes on the Greenland Ice Sheet. We acknowledge the importance of this prior research and its relevance to our study, which explores lake growth in response to exceeding zero-degree temperatures, utilizing Positive Degree Days (PDD) to assess lake evolution. We have included this work in our reference list as follows:

"This process is closely associated with the changes in temperature, especially the occurrence of days with temperatures exceeding 0°C (Braithwaite and Hughes, 2022; Li Qing, 2021; Wake and Marshall, 2015; Maisincho et al., 2014; Barrand et al., 2013; Johansson et al., 2013)." (lines 374-376)

**Presentation**

*There is a substantial amount of details about chemical and biological importance of these lakes in the introduction. Shorten this to one paragraph, up to 4 sentences and highlight instead how this work fits into lake detection using satellite images, ML/DL of lake detection, or similar. The work should be set into the context of existing science with the topic presented here not in a different scientific field.*

Thanks for your suggestions about the introduction. We have condensed the mentioned details into one concise paragraph, comprising up to four sentences as follows:

"Extensive research confirms diverse microorganisms in Antarctic lakes, including prokaryotes like bacteria and eukaryotes such as phytoplankton (Parnikoza and Kozeretska, 2019; Izaguirre et al., 2021; Keskitalo et al., 2013; Rochera and

Camacho, 2019). Cyanobacteria play a crucial role in primary production and nutrient cycling, as highlighted by studies on their diversity and distribution (Taton et al., 2006; Komárek et al., 2012), alongside findings on unique microbial assemblages, such as Hymenobacter sp., and diverse bacterial communities (Koo et al., 2014; Huang et al., 2014; Carvalho et al., 2008; Papale et al., 2017). These studies underscore the ecological importance and high diversity of Antarctic lake ecosystems." (lines 31-36)

*P7R145-150. If dual-polarization data is not available, why is it being discussed here where the method is presented? This would then fit better in the introduction or the discussion.*

Thanks for your suggestions. We moved this section to introduction as follows:

"In addition, compared to single-polarization SAR images, the utilization of multi-polarization SAR images can improve the capability to distinguish LLOW from other ground objects (Zakhvatkina et al., 2019). However, the high-resolution GRD products only provide single polarization over the Antarctic continent. The high-resolution multi-polarization SAR images are not available in Antarctica." (lines 66-69)

***Minor comments***

*P2R12 what is "reliable" in this context?*

We appreciate the reviewer's question concerning the use of the term "reliable" in the context of Antarctic landlocked lakes' open water (LLOW) serving as a climate indicator. In our manuscript, the term "reliable" is intended to convey the consistent and predictable nature of LLOW as an indicator of climatic conditions in the Antarctic ecosystem. LLOW's sensitivity to climatic variables, such as temperature and precipitation, enables it to reflect broader environmental changes. This sensitivity is based on the direct relationship between climatic factors and the physical, chemical, and biological processes occurring within these lakes, making them effective indicators of climate change.

*P2R16. Why did you choose ice-free areas? And what do you mean with ice-free here, no glacier ice, no inland ice sheet, no sea ice.*

Thank you for your question. The ice-free areas refer to the coastal Antarctic continental areas with no ice/snow in the austral summer; that is, no glacier ice, no inland ice sheet. In these ice-free areas, the lakes undergo freezing and melting cycles, which play a crucial role in maintaining the ecosystems of Antarctica. In addition, the changes in the areas of this type of lakes are sensitive to the climate warming. Thus, a thorough understanding of the areas of these lakes is of great significance to assessing the impact of climate change on Antarctic ecosystems.

However, the methods for detecting this kind of lakes are unavailable now. Thus, we aim to develop new techniques and methodologies to improve the detection and analysis of these lakes, contributing to the broader field of climate and ecological research in Antarctica.

*P4R63. Rapidly -> change to a more scientific wording.*

Thanks for your suggestion. We have revised "it can be rapidly applied to hundreds of satellite scenes" into "it can be automatically applied to hundreds of satellite scenes".

*Section 2.2. The number of optical images are introduced as 79 and then there is the discussion about removing image. It later transpires that 79 images were used. The text must be amended so that it is clear how many images were being used.*

Thanks for your comment. We used 79 Landsat images and 390 Sentinel-1 images at the beginning. We removed 60 ascending orbit images. Then in post-processing section, we removed 45 Sentinel-1 images which are affected by strong wind or other factors. Thus, we used a total of 285 Sentienl-1 images. We revised the manuscript to clarify the number of images in final products:

"Therefore, we disregarded those underestimated LLOW results and generated a total of 285 long term-series images of LLOW. The LLOW series combined Landsat and Sentinel images have a spatial resolution of 10 m and a time resolution of ~ 6 days." (lines 260-262)

*P5R103. Specify what "superior in many aspects" means.*

Thanks for your suggestion. We have revised this sentence as follows:

"The Landsat 8-9 OLI data are superior in the enhanced radiometric capabilities and the expanded range of spectral bands (Gorji et al., 2020)." (lines 98-99)

*P6R113. This is well known remove this reference to fundamental radar knowledge.*

Thanks for your suggestion. We have removed this reference.

*P6R117. Define high-resolution*

Thanks for your suggestion. We revised the manuscript as follows:

"To accurately determine the peak dates of landlocked lake area changes, the temporal resolution of area measurements needs to be at the weekly or daily time scale..." (lines 113-114)

*P6R125-127. Why is there a reference attached to one of the weather stations and not the other? Is it possible to give credit to the data providers instead? The text about "temperature" is confusing, is this not actual temperatures but some kind of simulated temperatures or why has been used?*

Thank you for your comments. Regarding the Davis station data, they are from the Australian Government Bureau of Meteorology's official website at http://www.bom.gov.au/climate/data/stations/, with the station number of 300000.

As for the term "temperature", refer to "daily mean air temperature" and "daily mean near-surface temperature". The quotation marks were used to emphasize this abbreviated expression. Please be assured that the station data represents actual observations and is not derived from reanalyzed data sources like ERA-5.

*Chapter 3. The ground truth should be presented in the data and not as a part of the results in chapter 3. This also goes for parts of chapter 3.2 that should also be moved to the data section.*

Thanks for your suggestion. We have moved description of annotated ground truth to chapter 2. Chapter 3.2 is about the process of open water identification, so it's not suitable for the data section. We revised the manuscript as follows:

"To train and validate the U-Net and the random forest (RF) model, we manually annotated ground truth labels from Landsat 8-9 OLI images and Sentinel-1 images. For the U-Net model, several Landsat images were selected, and the pixels in the images were annotated as "open water", "ice", and "rock" to serve as ground truth. To enhance the classification capability of U-Net in various scales, the side lengths of Landsat images ranged from 30 pixels to 200 pixels. We annotated 23 patches with 17100 pixels for U-Net. For the RF model, directly annotating the ground truth in SAR images is challenging and time-consuming, primarily due to their complex backscatter characteristics. Therefore, we conducted visual interpretation for Sentinel-1 images with the assistance of Landsat images (Liang and Liu, 2020). To ensure that the Landsat images represent the surface of Sentinel-1 images, we selected the Landsat and Sentinel-1 images with the closest dates. Due to the limited availability of cloud-free Landsat images, we used all cloud-free Landsat images from 2017 to 2021 to generate the sample "open water" and "others". In addition, to validate the accuracy of LLOW identification, we also annotated these Sentinel-1 images as "LLOW" and "others". We annotated 46 patches with the 300*300 size to train the RF model and validate the model accuracy. The 46 patches were randomly sampled at a 10% ratio to generate a sample point set. These points were then randomly divided into 80% for training and 20% for testing to train and test the RF model. Additionally, we identified the LLOW with the 46 patches and then calculated the accuracy, F1 score, and mean IoU to evaluate the identification accuracy." (lines 145-158)

*P11R226-229. Very well known (fundamental radar) remove reference.*

Thanks for your suggestion. We have removed this reference.

*Within this manuscript Sentinel-1 has been used, this is essential to call it or make an acronym if it's preferred to call it Sentinel. This as there are many ESA Sentinel satellites, and there is also the Asian Sentinels.*

Thanks for your suggestion. We will uniformly replace "Sentinel" with "Sentinel-1".

*P12R270. Many methods detect glacier outlines etc. A more thorough method should be able to at least attempt to separate ice (moving materiel) from the more stationary rocks.*

Thanks for your suggestion. We have revised the manuscript as follows:

[revised manuscript text omitted]